# EPicker is an exemplar-based continual learning approach for knowledge accumulation in cryoEM particle picking

Xinyu Zhang[1,2,9], Tianfang Zhao[1,2,9], Jiansheng Chen [3✉], Yuan Shen [1,2✉] & Xueming Li [4,5,6,7,8✉]

Deep learning is a popular method for facilitating particle picking in single-particle cryo-electron microscopy (cryo-EM), which is essential for developing automated processing pipelines. Most existing deep learning algorithms for particle picking rely on supervised learning where the features to be identified must be provided through a training procedure. However, the generalization performance of these algorithms on unseen datasets with different features is often unpredictable. In addition, while they perform well on the latest training datasets, these algorithms often fail to maintain the knowledge of old particles. Here, we report an exemplar-based continual learning approach, which can accumulate knowledge from the new dataset into the model by training an existing model on only a few new samples without catastrophic forgetting of old knowledge, implemented in a program called EPicker. Therefore, the ability of EPicker to identify bio-macromolecules can be expanded by continuously learning new knowledge during routine particle picking applications. Powered by the improved training strategy, EPicker is designed to pick not only protein particles but also general biological objects such as vesicles and fibers.

[1] Department of Electronic Engineering, Tsinghua University, Beijing 100084, China. [2] Beijing National Research Center for Information Science and Technology, Tsinghua University, Beijing 100084, China. [3] School of Computer and Communication Engineering, University of Science and Technology Beijing, Beijing 100083, China. [4] School of Life Sciences, Tsinghua University, Beijing 100084, China. [5] Tsinghua-Peking Joint Center for Life Sciences, Beijing 100084, China. [6] Beijing Frontier Research Center for Biological Structure, Beijing 100084, China. [7] Advanced Innovation Center for Structural Biology, Beijing 100084, China. [8] Key Laboratory for Protein Sciences of Ministry of Education, School of Life Sciences, Tsinghua University, Beijing 100084, China. [9]These authors contributed equally: Xinyu Zhang, Tianfang Zhao. ✉email: jschen@ustb.edu.cn; shenyuan_ee@tsinghua.edu.cn; lixueming@tsinghua.edu.cn

Single-particle cryo-electron microscopy (cryo-EM) is now a powerful tool for determining the atomic structure of bio-macromolecules in the solution. Single-particle cryo-EM processing involves a multistep workflow to obtain the structure by 3D reconstruction. Particle picking is a key step at the beginning of the workflow; it recognizes bio-macromolecular particles embedded in vitreous ice and determines their locations on micrographs. A cryoEM micrograph often contains multiple views of the target bio-macromolecules, degraded proteins, protein impurities, and ice contaminations. Particle picking is expected to precisely locate particles of protein complexes with a homogenous conformation and those with conformational or compositional heterogeneity. Owing to the requirement for efficiency of the cryoEM workflow, particle picking is also expected to be automated.

The basic concept of particle picking is to match given features to target images, which includes two steps: feature extraction and object detection. Traditional methods, such as FindEM[1], Signature[2], DoGpicker[3], gAutoMatch, and the picking sub-routines in EMAN[4] and RELION[5], are based on matching given templates or specific features. The user should explicitly prepare and provide template images or specific feature descriptions of the target samples. However, these methods suffer from the dependency of template preparation, which often strongly depends on the user's experience and can easily cause bias. As an alternative to template matching, several algorithms that do not require the user to provide templates have been developed, such as DeepCryoPicker[6] and DRPNet[7], which automatically obtain features using clustering and unsupervised learning algorithms, respectively. In recent years, deep-learning-based methods, especially convolutional neural networks (CNNs), have shown great potential for particle picking. CNNs are more adaptable and automated than traditional methods. Rather than using intuitive and visible features, deep learning algorithms can automatically learn to extract abstract and hierarchical features from labeled samples via a multi-layer neural network and generate a parametric model. This process is called training. Then, based on the model, particle picking can be performed through the inference process.

Wang et al. employed a CNN in DeepPicker[8] for particle picking. By joint training on several datasets of diversified protein complexes, DeepPicker demonstrated the generalization capability of CNNs in the task of particle picking. Later, more particle picking approaches and programs, such as DeepEM[9], Warp[10], Topaz[11,12], and crYOLO[13], used CNN or modified CNN. In these methods, joint training is applied to multiple datasets. For instance, crYOLO[13] used 53 datasets; the developer of Warp[10] suggested a central repository of training data and periodic training. Frequently adding new features is necessary to broaden the applicable range of a general model. However, joint training on an increasing number of datasets is computationally intensive and requires a large storage space. Alternatively, fine-tuning[14] is used to quickly adapt to unseen features. However, fine-tuning can only generate a specific feature model rather than a general model, that is, the new model loses its ability to effectively pick old particles, known as catastrophic forgetting[15].

A low-cost alternative to joint training is continual (or incremental) learning, which aims to adapt a new model for a new task while maintaining performance for old tasks[16–24]. Knowledge distillation[25] is a widely used technique for incremental object detection problems in natural images[26–28], which transfers knowledge between neural networks using loss functions that minimize the difference between features extracted from the new and old models for old datasets. Continual learning enables the accumulation of knowledge in existing models. Hence, with an increasing amount of incorporated data, continual learning should be able to continuously enhance particle picking in a cryoEM pipeline.

Here, we report an EPicker program with an exemplar-based continual learning algorithm based on a CenterNet object detector[29] for particle picking in cryoEM. EPicker is shown to enhance the performance of particle picking continuously with more new knowledge of the features learned. EPicker also supports joint training and fine-tuning to meet the different requirements of particle picking. The characteristics and possible uses of different training modes are discussed. EPicker is designed to pick general biological objects, including protein particles, liposome vesicles, and fibers. All these features make EPicker highly advantageous for both automated cryoEM pipelines and single-user applications.

## Results

**Continual learning for accumulating knowledge of features.** A model for comprehending the features of bio-macromolecules using a deep learning approach is the basis of cryoEM particle picking and determines the particle picking performance. We implemented an exemplar-based continual learning approach in EPicker to enhance its ability to adapt to new features. Continual learning refers to the gradual addition of new knowledge to an old model through further training with new datasets. The exemplar is used to guide continual learning to avoid forgetting old knowledge when learning new knowledge (discussed later). EPicker uses a CenterNet detector[29] as the basic network. However, CenterNet alone does not support continual learning. We designed a dual-path network for continual learning in EPicker based on CenterNet. Theoretically, similar networks can also be adopted in the dual-path architecture to enable continual learning.

In the dual-path network of EPicker, the two paths have the same network structures, referred to as branches A and B, as shown in Fig. 1. Under the configuration of CenterNet, each branch is composed of a feature extraction sub-network and an object location sub-network. In the training process, both branches are initialized using the same parameters as in the old model. Branch A is fixed during training and is considered as a reference for old features, which preserves the old knowledge. Branch B is used to generate the new model by distilling the knowledge from branch A to avoid catastrophic forgetting. Based on this design, the parameters of the old model initially loaded into branch B are iteratively updated on the exemplar dataset and the new dataset. The exemplar dataset is a subset of the datasets used for training the old model. Empirically, an exemplar dataset contains approximately 200 labeled particles (distributed on one or multiple micrographs) for each particle dataset. Random flip and random cropping are used for data augmentation to improve the generalization ability of the network. After the training process, EPicker discards branch A and saves only the parameters of branch B as the new model.

A loss function with three components was designed to determine the update of branch B (see Methods). An exemplar dataset of the old model was input into the two branches to extract the features. A knowledge distillation loss function, $L_{Distill}$, was used as a constraint to minimize the difference between the extracted features and generated heatmaps from branches A and B. Joint training on the exemplar dataset and the new dataset was conducted with an object detection loss function, $L_{OD}$, on branch B. The purpose of using the exemplar dataset is to avoid the gradient descent direction along the gradient of the new dataset (red arrow in Fig. 1). The latter is the reason for catastrophic forgetting. The joint training on the two datasets combined their gradient descent directions by balancing the loss on both the new

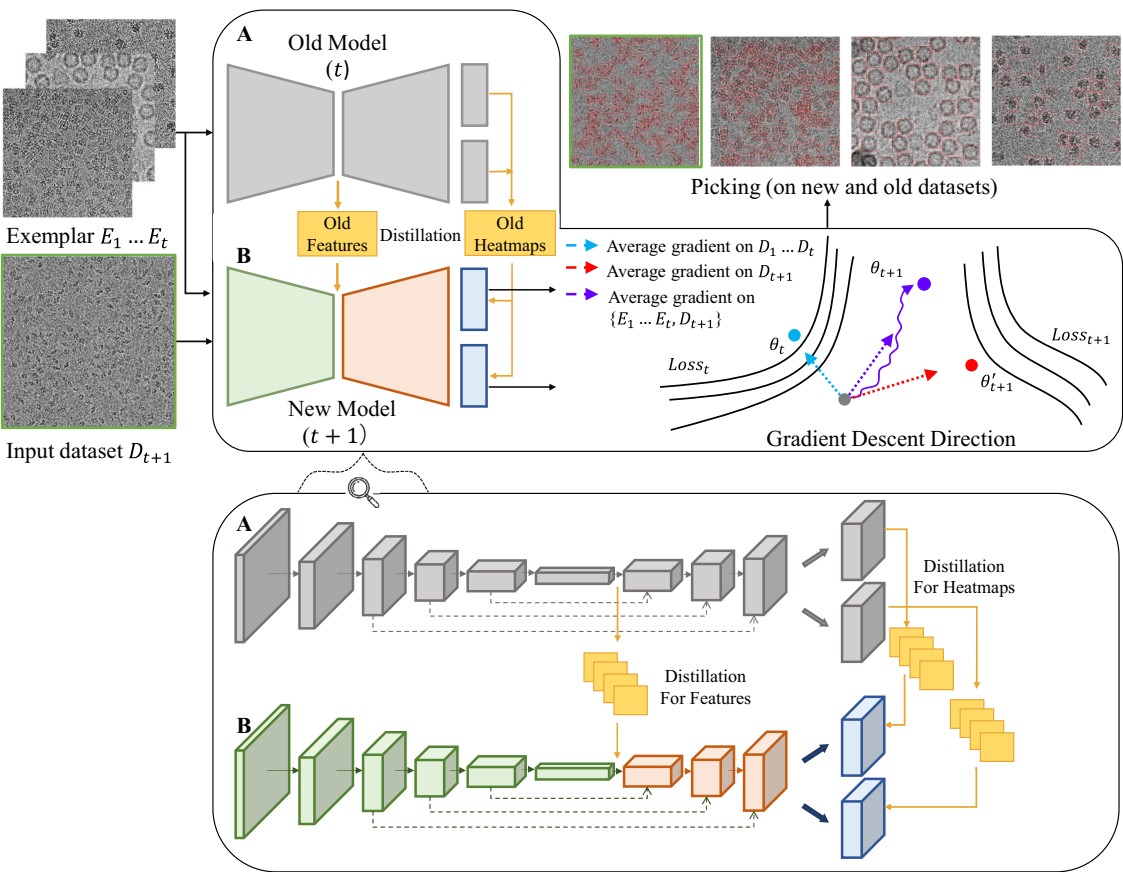

**Fig. 1 Architecture of EPicker and schematic diagram of continual learning.** The input data includes an exemplar dataset of datasets $E_1$--$E_t$ and a new dataset $D_{t+1}$ (highlighted by green boxes). A and B represent branches A and B of the dual path network, respectively. Each branch is composed of a feature extraction sub-network (two trapezoids) and an object location sub-network (two rectangles). The gradient descent direction on the exemplar dataset $\theta_t$, the new dataset $\theta'_{t+1}$, and the combination of the two datasets $\theta_{t+1}$ is indicated by blue, red, and purple arrows, respectively. The bottom panel shows the magnified details of the network, in which branch B distills knowledge of features and heatmaps (yellow squares) from the corresponding network blocks (cuboid) of branch A.

and old datasets. Third, to avoid over-fitting on the exemplar dataset, a regularization loss function, $L_{Reg}$, is calculated as a measure of the difference between the new and old models, which penalizes significant changes between the parameters of the new and old models. Finally, the total loss is calculated as follows:

$$L_{Total} = L_{OD} + \lambda_d * L_{Distill} + \lambda_r * L_{Reg}, \qquad (1)$$

where $\lambda_d$ and $\lambda_r$ are hyper-parameters used to balance the importance of the corresponding loss terms. In all our experiments, we empirically set $\lambda_d = 0.1$ and $\lambda_r = 0.01$ (Supplementary Table 1).

The impact of each component in the loss function was evaluated (Supplementary Table 2), which demonstrated that a combination of the three components is necessary. We also compared the performance of the proposed method with that of other widely used incremental learning methods (see Methods, Supplementary Table 3). The exemplar-based method in EPicker exhibited the best stability on the evaluation datasets. The exemplar dataset and continual learning process of EPicker mimic human behavior. If the old model is imagined as a memory of the past, the exemplar dataset is a note and snapshot of past events.

**CenterNet detector in EPicker.** CenterNet[29] was used as the basic network for particle picking in EPicker. CenterNet is an one-stage anchor-free object detection network based on keypoint

detection, and has shown better performance than existing anchor-based methods, such as YOLO[30–32], RetinaNet[33], and Faster R-CNN[34]. The CenterNet detector can regress both the position and size of the object and is thus suitable for particle picking in cryoEM. Several feature extraction networks are available for CenterNet, such as ResNet[35] and DLA[36]. Compared with ResNet, DLA improves the ability of feature representation by adding more skip connections and exhibits better performance for the particle picking task (see Methods, Supplementary Table 4). Hence, EPicker chose a DLA network with 34 convolutional layers (DLA-34) as the feature extraction sub-network (Fig. 2). The feature extraction network extracts feature maps from the input micrographs using a series of convolution operations and subsequent deconvolution operations. Then, the object location network processes the feature maps to generate heatmaps to predict the position and size of the particles. Each branch of the dual-path network for continual learning takes the same network configuration as described here.

**Optimizations for the network settings.** In addition to the network structure optimization described earlier, several additional settings were used to improve the overall performance of EPicker. Considering that particles of the same protein sample are homogenous in size, EPicker turns off the prediction of particle size, that is, it only regresses the particle position in the object location sub-network, which helps to reduce computing complexity and improve position estimation accuracy. For size-

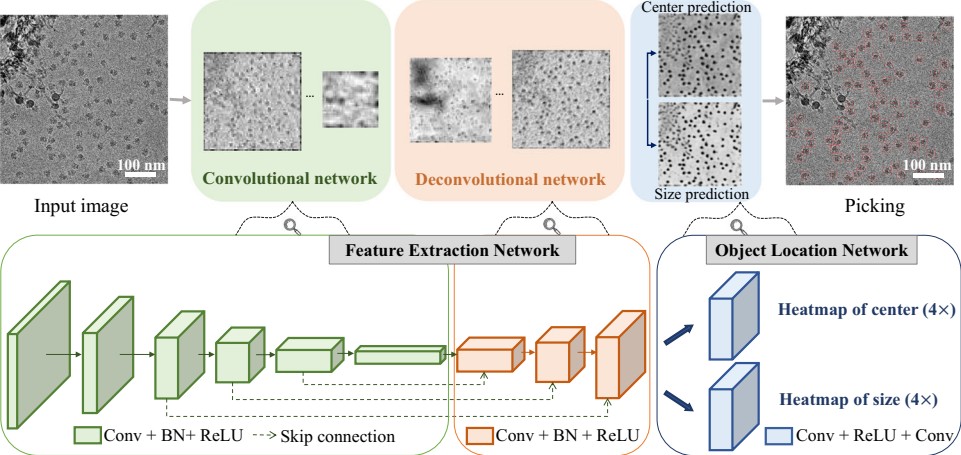

**Fig. 2 Architecture of CenterNet detector in EPicker.** The feature extraction sub-network is a cascade of a convolutional network (green) and a deconvolutional network (red) to extract features. Both the convolutional and the deconvolutional networks are a combination of Convolution-Batch Normalization[46]-ReLU(Rectified Linear Unit)[47] blocks. The object location sub-network (blue) of the detector generates the heatmaps of particle center and size. The heatmap regression network is a combination of Convolution-ReLU-Convolution blocks.

sensitive cases, such as liposomes (discussed later), size estimation can be turned on to output the particle radius together with position (Supplementary Fig. 1). To accelerate the computation, the input micrograph is down-sampled to a fixed width of 1024 pixels and a correspondingly scaled height to maintain the aspect ratio. For a typical particle size of 10–30 nm, particle picking is usually tolerant to a centering error of less than 1 nm, corresponding to 4–5 times the pixel size of the micrographs. Therefore, the reduced micrograph size does not have an obvious influence on the position accuracy. The scaled micrograph is then processed by histogram equalization and converted to an 8-bit format with 256 gray levels. The reduced image size accelerates the particle picking, typically, to less than 0.3 s for one micrograph (Supplementary Table 5).

**Training by continuously adding new datasets**. The continual learning method implemented in EPicker supports the addition of one or a group of datasets during the training, which allows for the gradual enhancement of the automated pipeline system. We evaluated the picking performance by incrementally adding datasets to mimic the activities of a gradually enhanced system. The performance of EPicker was evaluated using average precision (AP) and average recall (AR) under a given threshold 0.5 of intersection over union (IoU, see Methods).

To demonstrate the reliability and robustness of continual learning on datasets with different features, we included particles as diverse as possible, considering the structural features, shape, and size (Supplementary Table 7). For the experiments, a basic model was first obtained by joint training on five datasets including 80S ribosome (EMPIAR-10028), 20S proteasome (EMPIAR-10025), apoferritin (EMPIAR-10146), TccA1 (EMPIAR-10089), and Nodavirus (EMPIAR-10203). Five new datasets, β-galactosidase (EMPIAR-10017), influenza hemagglutinin (EMPIAR-10097), phage MS2 (EMPIAR-10075), CNG (EMPIAR-10081), and phosphodiesterase (EMPIAR-10228) were used individually or in groups (Table 1) for further training. From each dataset, we selected 15 micrographs, of which 10 were used as the training dataset and five as the test dataset (Supplementary Table 6). All particles were manually picked and used as ground truth.

Joint training was first conducted on all the aforementioned 10 training datasets. While these molecules have very different structural features, sizes, and molecular weights from 100 kDa to several MDa, the picking performance is maintained at a high level (Table 1),

which reflects the great generalization capability of the feature extraction sub-network. Generalization is the basis for adding new knowledge to an existing model. The picking results based on the joint-training model are considered to be the upper bound of the performance of the EPicker.

Continual learning was then performed based on the basic model of five datasets. Adding new datasets in a continual manner caused a 1–3 % decrease in AP value and little influence on AR value compared with the corresponding results of the joint-training model. Adding a group of five new datasets together was also evaluated, demonstrating nearly the same influence as adding the datasets successively. The dissimilarity of features between the old and new datasets may influence the effectiveness of merging different features, which is indicated by forgetting some old features. To measure this influence, we defined the complexity of a new dataset as how the features in the new dataset match the features in the old datasets and defined the forgetting rate as the reduction of AP and AR (see Methods). We then evaluated the relationship between complexity and forgetting rate (Supplementary Fig. 2). The experimental results show that adding new datasets with different features does not cause significant forgetting. Meanwhile, adding datasets with low complexity can improve the model. Therefore, the picking performance of the new model in picking old samples should be maintained and nearly not influenced after learning more features.

Moreover, the continual learning ability of EPicker significantly reduced both the time and storage costs of extending new features (Supplementary Fig. 3). The time spent on a single joint training with 5–10 datasets was 26–50 min and increased linearly with more datasets involved. For joint training, once a new dataset was added, the training was performed repeatedly on all previously involved datasets. All complete training datasets (10 micrographs per dataset for the current experiments) should be stored for future training. In contrast, for the continual learning process, the time to add a new dataset is usually significantly less than joint training and increases slightly with the accumulation of more features. In the situation of gradually adding new datasets, the joint training on adding the 10th dataset costs 50 min, while continual learning takes only 20 min. What's more, only a small number of samples (typically 1–2 micrographs) in each dataset need to be stored in the exemplar dataset for future continual learning.

**Catastrophic forgetting in fine-tuning**. In EPicker, the fine-tuning and joint training modes adopt the same training method

**Table 1 Evaluation of particle picking (AP/AR) under different training modes.**

| Name | E10089 | E10146 | E10028 | E10203 | E10025 | E10017 | E10097 | E10075 | E10081 | E10228 | mAP/mAR |
|---|---|---|---|---|---|---|---|---|---|---|---|
| NO. | (1) | (2) | (3) | (4) | (5) | (6) | (7) | (8) | (9) | (10) | |
| JT($D_1$–$D_{10}$) | 97.1/99.1 | 96.3/97.2 | 96.8/97.5 | 93.5/98.3 | 92.1/96.7 | 96.8/99.4 | 92.7/97.2 | 96.0/96.4 | 95.3/97.6 | 92.5/97.4 | 94.9/97.7 |
| JT($D_1$–$D_5$) | 97.5/99.1 | 96.4/97.2 | 96.7/97.5 | 93.1/98.9 | 92.0/96.6 | **19.8/50.2** | **9.7/64.9** | **95.1/98.7** | **80.9/92.3** | **26.5/77.6** | 95.1/97.9 |
| JT($D_1$–$D_6$) | 97.5/99.1 | 96.8/97.5 | 97.6/98.0 | 95.3/99.4 | 92.0/96.7 | 97.0/99.5 | **19.6/77.9** | **95.0/96.7** | **83.4/95.2** | **37.0/86.9** | 96.0/98.4 |
| FT($D_6$) | 35.8/97.5 | 89.9/95.9 | 67.1/97.7 | 69.2/98.3 | 68.4/87.6 | 96.6/99.0 | 61.6/96.5 | 27.0/94.4 | 51.6/97.7 | 36.8/95.9 | 71.2/96.0 |
| CL | 96.4/99.2 | 94.9/97.2 | 96.4/97.7 | 90.2/98.3 | 90.3/95.5 | 96.2/98.5 | 66.5/97.2 | 92.2/96.7 | 84.4/95.2 | 55.9/97.0 | 94.1/97.7 |
| ($E_t$–$E_t$ + $D_{t+1}$) | 96.1/98.4 | 96.2/97.6 | 96.6/97.3 | 89.4/97.7 | 90.2/95.7 | 96.4/98.8 | 93.9/98.5 | 92.2/96.4 | 88.6/96.9 | 84.1/97.4 | 94.1/97.7 |
| $t$ = 5–9 | 96.1/98.8 | 96.7/97.9 | 96.7/97.3 | 92.6/98.9 | 89.9/95.3 | 96.4/99.0 | 91.9/96.8 | 95.7/96.7 | 80.1/95.0 | 83.4/96.4 | 94.5/97.6 |
| | 96.4/98.8 | 96.7/97.7 | 96.7/97.3 | 91.9/98.9 | 89.7/95.8 | 95.8/98.3 | 92.8/97.5 | 96.4/97.1 | 94.9/97.7 | 80.8/96.8 | 94.6/97.7 |
| | 96.4/98.8 | 96.8/97.6 | 96.6/97.3 | 90.1/98.9 | 89.6/95.6 | 94.6/97.9 | 91.8/97.5 | 95.8/96.7 | 93.1/96.9 | 91.8/97.2 | 93.7/97.4 |
| CL ($E_1$–$E_5$ + $D_6$–$D_{10}$) | 96.2/98.3 | 96.2/97.9 | 96.6/97.3 | 91.1/98.3 | 91.4/96.3 | 97.6/99.4 | 91.5/97.0 | 96.1/97.7 | 94.9/97.4 | 93.8/98.0 | 94.5/97.7 |

The 10 datasets are indicated by $D_1$–$D_{10}$, the corresponding exemplar dataset are indicated by $E_1$–$E_{10}$. The joint training (JT), fine-tuning (FT), and continual learning (CL) are tested with different datasets. The cells with bold values indicate that the corresponding dataset is unseen for the model. The cells with underlined values indicate that the corresponding dataset is newly added for the model. mAP and mAR indicate the mean of the average precision (AP) and the mean of the average recall (AR), respectively, of all columns in the corresponding row.

and fine-tuning suffers from catastrophic forgetting problem. The difference between fine-tuning and joint training lies in whether to load a pre-trained model, that is, the former loads the parameters of an existing model and trains on one or multiple new datasets, whereas the latter trains on a combination of datasets from scratch. The network used for fine-tuning and joint training uses a single-path network compared with the dual-path network used for continual learning. The single-path network has the same structure as one branch of the continual-learning dual-path network and lacks a reference network for maintaining the old knowledge. During the fine-tuning process, the parameters of all network layers were greatly modified to ensure better performance on the new datasets. The fine-tuning model cannot extract and maintain old features from the old model.

Based on the basic model jointly trained on the five datasets mentioned in the previous section, we compared the fine-tuning mode with the continual learning mode on the β-galactosidase (EMPIAR-10017) dataset. A joint training model trained on all six datasets was also compared. Using the three new models, we selected a typical micrograph with 118 particles in the 80S ribosome dataset (EMPIAR-10028, considered as a dataset that appeared in the previous training) (Fig. 3a). The picking using the fine-tuning model missed 35% of the ground-truth particles (Fig. 3d), demonstrating catastrophic forgetting. The continual-learning and joint-training models pick 96–97% of the ground-truth particles (Fig. 3b, c). The fine-tuning model only achieved high performance on the new β-galactosidase dataset, and the other two models worked well on all six datasets (3rd–5th rows in Table 1). Therefore, fine-tuning generates a specific model and forgets some old knowledge.

**Biased and unbiased picking.** Both continual learning and joint training based on a large number of datasets are unbiased to particles with different features and tend to generate a general model for extracting various features. In contrast, fine-tuning generates a model that is specific or biased to the features of the latest training datasets. Unbiased picking is often important, especially at the beginning of a new project, to find as many particles with unknown conformations or compositions as possible. Sometimes biased picking is also necessary, mostly to quickly find particles with specific features in order to improve the resolution of 3D reconstruction. EPicker supports three training modes to satisfy the requirements for different picking specifications.

We used micrographs of the 26S proteasome with mixed assembly states to test the picking specificity of the models from different training modes. The 26S proteasome (26S) is composed of a 20S core particle (CP) and two 19S regulatory particles (RP) that bind to CP. Because RP can disassemble from 26S, different complexes are often observed, including stand-alone CP, CP with two RPs (CP2RP), and CP with one RP (CP1RP). The diversity of the complexes provides an opportunity to test picking specificity. In the experiment, we regarded the side-view CP2RP as the positive sample; accordingly, all other impurities and complexes in different assembly states (CP and CP1RP) were considered as unknown or junk particles.

Before working on 26S, we assumed that 26S was unknown, and our general model was trained on 46 datasets without any 26S or its components. First, particles were picked from 200 micrographs in a dataset with a high-purity side-view 26S (EMPIAR-10090) using the general model. As expected, only a small number of CP2RP particles were picked because the general model has never seen 26S. Then, by performing a 2D classification on the poorly picked 26S particles (Supplementary Fig. 4), some side-view CP2RP particles were selected and used to

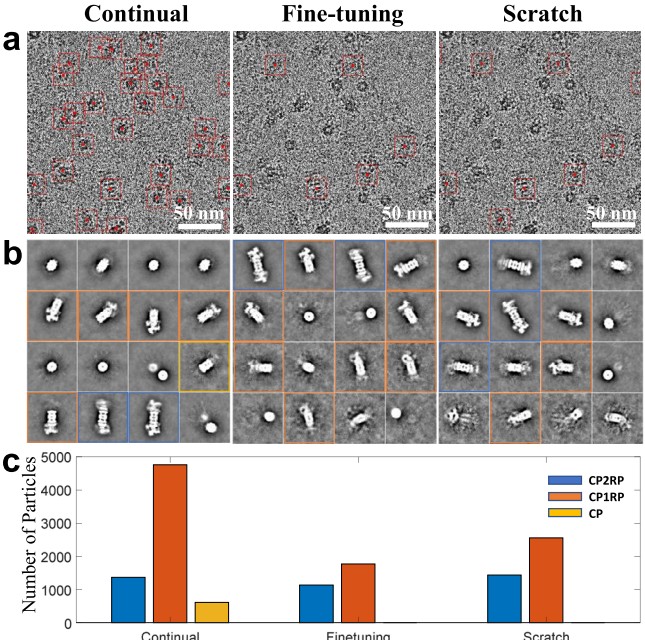

**Fig. 3 Illustration of the catastrophic forgetting.** The picked particles of 80S ribosome (EMPIAR-10028) are annotated by square boxes. The red, blue, and yellow boxes indicate the particles were correctly picked, wrongly picked, and missed, respectively. **a** The ground truth. **b** The results of the joint training model. **c** The results of the continual learning model. **d** The results of the fine-tuning model.

**Fig. 4 Comparison of biased and unbiased picking on a 26S proteasome dataset. a** The particle picking results (red boxes) of the three models. To improve the visualization, a small region of a typical micrograph is shown. The continual-learning model results in the picking of nearly all particles appearing on the micrograph. The results using the fine-tuning and scratch models are more specific on the side-view of CP2RP (a core particle with two regulatory particles) particles and ignore other particles. **b** Part of the 2D (two-dimensional) class averages with the highest occupancy, which are sorted in descending order of the occupancy. Different border colors specify CP2RP (blue), CP1RP (a core particle with one regulatory particles, orange), and CP (core particle, yellow) particles, respectively. In the results of the fine-tuning model, the CP2RP classes have higher occupancy than other particles. **c** Plot of the total number of particles picked with different models. The three models picked nearly the same number of CP2RP particles (blue bars). The fine-tuning and scratch model missed many CP1RP particles (orange bars) and nearly all CP particles (yellow bars), indicating biased picking.

train a general model in continual learning mode, a specific model in fine-tuning mode, and a model from scratch in joint training mode (simply called scratch model).

To compare the specificity of picking, the three models were applied to another 26S dataset (EMPIAR-10401) with many disassembled particles and much lower contrast. The continual learning model picked nearly all particles on the micrographs, as expected (Fig. 4a), including the circular top-view particles of 26S and the side-view particles of CP2RP, CP1RP, and CP. The

picking by the fine-tuning model and the scratch model are more specific, mostly focused on the side-view CP2RP particles and a small number of side-view CP1RP particles (Fig. 4a). Further 2D classification analysis (Fig. 4b) showed that the three models picked nearly the same number of CP2RP particles (Fig. 4c), whereas the fine-tuning model was the most accurate for picking the specific CP2RP particles (Fig. 4b). The fine-tuning and scratch model missed many CP1RP particles and nearly all stand-alone CP particles (Fig. 4c).

The aforementioned comparison provides a detailed insight into the different training modes and their behaviors on unseen particles. The general model is powerful in picking unseen particles and can be efficiently enhanced by incorporating more knowledge through continual learning.

**General object picking for cryoEM**. Benefiting from the capability of continual learning, EPicker has the potential to pick more general objects by accumulating more knowledge during long-term applications. Currently, EPicker supports picking for three types of objects with different features, not only the aforementioned particles but also fibers and vesicles. The fibers were processed as a series of discrete points in EPicker. The picking and training algorithm for the fibers was the same as that for the particles. An internal algorithm (see Methods) was developed to link the picked points as lines tracing the fibers. EPicker can deal with both curved and straight fibers (Fig. 5a, Supplementary Fig. 5b). Vesicles are usually liposomes that have recently become popular in the study of membrane proteins[37]. Considering that some membrane proteins are sensitive to the curvature of membrane bilayers, that is, the radius of the liposome, EPicker predicts the size of vesicles (Supplementary Fig. 1). Tests on liposomes showed that EPicker can accurately estimate both the center coordinate and size of each vesicle, even for overlapped vesicles (Fig. 5b).

EPicker is suitable for cryo-EM pipelines with continuous data input (Fig. 5c). Particle coordinates from 2D/3D classification of single-particle analysis and manual picking, which can be sparse (see Methods), are available for training. Pre-trained general models are available together with EPicker through the website http://thuem.net.

## Discussion

Particle picking is an important step in identifying biomacromolecules preceding 3D reconstruction in the cryoEM processing workflow. Any particles missed during particle picking are excluded in the final 3D reconstruction, which affects the reconstruction of molecules with unknown features. The level of matching between the knowledge accumulated in a network model and the features of known or unseen protein particles are vital for particle picking. While a general model can be used to pick unseen particles, as in the cases of the EMPIAR-10090

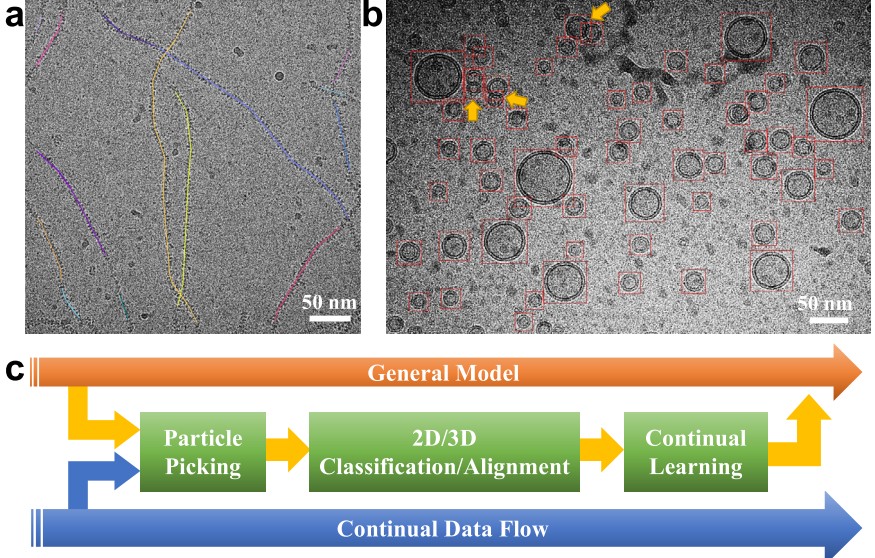

**Fig. 5 Picking of general biological objects and a workflow with continual learning. a** Picking curved fibers. The picked fibers are labeled as lines tracing the fibers. **b** Picking liposomes with estimation of radius (indicated by size of the red box). Several overlapped vesicles (indicated by yellow arrows) are well identified. **c** A cryoEM processing workflow with continual learning. Particle annotations can be obtained from manual picking, sparse picking, and 2D/3D (two-dimensional/three dimensional) classification, which can be further used to train the model in a continual manner.

dataset (Supplementary Fig. 4a) and the Fab dataset (the antigen binding fragment, ~60 kDa, Supplementary Fig. 6a), its performance is often not guaranteed.

We implemented a continual learning algorithm in EPicker to enhance the ability of feature extraction and generalization on more different particles, which is an efficient and convenient way to accumulate knowledge into an existing model. EPicker adopted a dual-path network to support knowledge distillation, used an exemplar dataset, and a specially designed loss function to avoid catastrophic forgetting.

Through experiments that mimicked real-world applications by gradually adding new datasets, the continual learning algorithm in EPicker successfully accumulated knowledge into an existing model, regardless of the complexity of the dataset; meanwhile, no obvious catastrophic forgetting was observed. We further compared the influence of several training modes for particle picking, including continual learning, joint training, and fine-tuning. The fine-tuning model was specific and biased to given features and suffered from catastrophic forgetting. The continual learning model achieved a performance similar to that of the joint training model. Furthermore, the continual learning strategy disperses the computationally intensive training on a large number of datasets into multiple training processes and thus it is not necessary to finish the training at one time as required by the joint training.

Enhanced by improving the capability to learn new features, EPicker is designed to pick more general objects in cryo-EM micrographs, including fibers and vesicles. To further improve the convenience of training, EPicker only requires positive annotations and supports sparse annotations by a specially designed loss function, that is, only a small part of the particles in the micrographs need to be annotated for training. Moreover, 5–10 micrographs from a dataset are usually sufficient for training. Based on these features, a small number of manually picked particles or particles selected by the 2D/3D classification can be used to build the training datasets. Finally, all these features make EPicker more reliable (Supplementary Fig. 6), easy to use, and suitable for most of the current requirements of object detection in single-particle cryoEM and automated workflow.

## Methods

**Continual learning algorithm**. EPicker is an exemplar-based incremental particle picking program. Guided by the exemplar dataset, EPicker can be trained on the new dataset without forgetting old knowledge. EPicker uses CenterNet[29] for object detection and the object detection process can be seen as a function, denoted by $F_{OD}$. The CenterNet detector in EPicker (Fig. 2) consists of two components: a feature extraction sub-network, $F_{extract}$, which extracts the features of the input image, and an object location sub-network, $F_{loc}$, which regresses the center, local offset, and size of each particle. Let $(D_1 \dots D_t)$ denote a set of old datasets sequentially added until time $t$. From each old dataset $D_i$ we randomly choose 200 continuous annotations on one or multiple downsampled micrographs $E_i \in D_i$. All of these micrographs comprise an exemplar dataset $(E_1 \dots E_t)$. Assuming that at time $t$, the object detector $F_{OD}^t$ is parameterized by $\theta_t$, EPicker incrementally adapts the parameter $\theta_t$ toward $\theta_{t+1}$ by training on a new dataset $D_{t+1}$ with the guidance of the exemplar dataset $(E_1 \dots E_t)$.

Figure 1 represents the involvement of exemplar in the training process. We use $Loss_t$ to represent the loss function of the network trained on old datasets $(D_1 \dots D_t)$ and $Loss_{t+1}$ to represent the loss function of the network trained on the new datasets $D_{t+1}$. When the network is trained on $(D_1 \dots D_t)$, the gradient descent will be along the direction where the loss function decreases fastest for the old datasets (indicated by the blue arrow in Fig. 1). When the network is trained on $D_{t+1}$, the gradient descent follows the optimal direction for the new dataset (indicated by the red arrow in Fig. 1). Without the use of the exemplar, the optimal parameters of the network trained on $D_{t+1}$ are $\theta'_{t+1}$, which often incurs a large loss on the old datasets, thus leading to catastrophic forgetting on the old datasets.

To mitigate the problem of forgetting, the exemplar $(E_1 \dots E_t)$ that contains a set of images from each old dataset is introduced to the network; thus, EPicker can integrate the gradient information of the old datasets into the new model. When the model is training on the new dataset $D_{t+1}$, the instructive exemplar influences the gradient descent direction. Then, the final average gradient descent direction can be adjusted to a more appropriate direction (indicated by the purple arrow in Fig. 1), and the parameters of the model reach $\theta_{t+1}$, which balances the performance on all datasets. The exemplar dataset is advantageous in retaining knowledge from the old model to the new one. EPicker adopts the knowledge distillation method[25] to constrain the parameters of the new model by maintaining a certain similarity to the old model, and also finds the optimal parameters for the new datasets. In contrast to regularization methods[16,17,21] that penalize the change of important parameters of the old model, EPicker treats each parameter equally and only constrains the discrepancy between the output feature maps of the new and the old models, which reduces the probability of having conflicting parameters between different tasks.

The corresponding loss functions used in the above process are discussed in the following two sections in detail.

**CenterNet detector and object detection loss function**. EPicker uses a CenterNet[29] detector as the basic object detection framework to predict the central coordinate and particle size. There are many choices for the feature extraction

network in CenterNet, such as ResNet[35] and DLA[36]. ResNet introduces a residual block to stabilize the training process and extract hierarchical features. The DLA network aggregates feature representations and fuses information across different layers by adding hierarchical and iterative skip connections. The DLA-based feature extraction network achieved better object detection accuracy than ResNet in our particle picking tests (Supplementary Table 4). We chose a fully convolutional upsampling version of DLA-34[36] as the feature extraction network of CenterNet in EPicker and followed by an object location sub-nextwork[29] to predict the center $\hat{Y}$, local offset, $\hat{O}$, and size $\hat{S}$. The final particle position is the sum of the center and offset. The object location network is composed of three convolution networks that generate downsampled heatmaps for the center, the local offset, and the size of each particle. Then, particle centers are predicted from a heatmap matrix, $\hat{Y} \in [0,1]^{\frac{W}{R} \times \frac{H}{R}}$, where $W$ and $H$ represent the width and height of the input image, respectively, and $R$ is the output stride after a series of convolution operations and is set to 4 in EPicker. Each cell of the heatmap $\hat{Y}$ records a score between [0, 1] to present the detection confidence, which is known as the confidence score. A higher confidence score indicates a higher probability of a particle being in the current cell. The peaks in the heatmap were predicted as particle centers $(\hat{x}, \hat{y})$. The offset prediction, $\hat{O} = (\delta \hat{x}, \delta \hat{y})$, and the size prediction, $\hat{S} = (\hat{w}, \hat{h})$, together with the predicted particle centers, determine the bounding box of a particle represented as $\left(\hat{x} + \delta \hat{x}, \hat{y} + \delta \hat{y}, \frac{\hat{w}}{2}, \frac{\hat{h}}{2}\right)$.

The loss function of the object detector[29] without continual learning strategies is defined as

$$L_{OD} = L_k + \lambda_{off} * L_{off} + \lambda_{size} * L_{size}, \tag{2}$$

where $\lambda_{off}$ and $\lambda_{size}$ are weighting factors, and $\lambda_{off} = 1$, $\lambda_{size} = 0.1$ are used in EPicker (Supplementary Table 8). The definitions of the three components are as follows.

- $L_k$ is a pixel-wise focal loss[33] that reduces the error between the predicted particle center, $\hat{Y}_{xy}$, and the ground truth particle center, $Y_{xy}$; $L_k$ is defined as

$$L_k = -\frac{1}{N}\sum_{xy}\begin{cases} (1 - \hat{Y}_{xy})^\alpha \log(\hat{Y}_{xy}) & if\ Y_{xy} = 1 \\ (1 - Y_{xy})^\beta (\hat{Y}_{xy})^\alpha \log(1 - \hat{Y}_{xy}) & otherwise \end{cases}, \tag{3}$$

where $N$ is the number of particle in the input image, $\alpha$ and $\beta$ are two hyperparameters, and $\alpha = 2$ and $\beta = 4$ are used in EPicker.

- $L_{off}$ is the loss of the local offset for each particle center caused by the output stride and is defined as

$$L_{off} = \frac{1}{N}\sum_{p}\left| \hat{O}_{\tilde{p}} - \left(\frac{p}{R} - \tilde{p}\right)\right|, \tag{4}$$

where $R$ represents the output stride, $\hat{O}_{\tilde{p}}$ represents the predicted offset at the low-resolution center point $\tilde{p}$, and $(\frac{p}{R} - \tilde{p})$ represents the ground truth center offset.

- $L_{size}$ is the loss of the particle size, which is optional in EPicker, and is defined as

$$L_{size} = \frac{1}{N}\sum_{k=1}^{N}|\hat{S}_{p_k} - s_k|, \tag{5}$$

where $\hat{S}_{p_k}$ represents the predicted particle size at the center point $p_k$, and $s_k$ represents the ground truth size of particle $k$.

**Loss functions used in different training modes**. EPicker supports three training modes: joint training, fine-tuning, and continual learning. The loss functions of joint training and fine-tuning are the same and consist of only one object detection loss term, given by

$$L_{Total} = L_{OD}. \tag{6}$$

The loss function of continual learning consists of three loss terms, given by

$$L_{Total} = L_{OD} + \lambda_d * L_{Distill} + \lambda_r * L_{Reg}, \tag{7}$$

where $\lambda_d$ and $\lambda_r$ are hyperparameters. EPicker empirically sets $\lambda_d = 0.1$ and $\lambda_r = 0.01$ in all experiments (Supplementary Table 1). The details of each loss term are discussed below.

- The object-detection loss $L_{OD}$, defined in Eq. 2 minimizes the particle center location error, offset regression error, and size regression error based on the ground truth and the prediction from the object location network, $F_{loc}$.
- The knowledge distillation loss, $L_{Distill}$, distills features generated by the feature extraction network, and the particle position heatmaps predicted by the object location network from the old model. For continual learning on model $t + 1$, EPicker only distills knowledge on the exemplar ($E_1 \dots E_t$) and does not involve that on the new dataset $D_{t+1}$. We use $L_2$ loss for

knowledge distillation, which is formulated as follows:

$$L_{Distill} = \frac{1}{M_f}\sum\|f_{t+1} - f_t\|_2^2 + \frac{1}{M_y}\sum\|y_{t+1} - y_t\|_2^2 + \frac{1}{M_o}\sum\|o_{t+1} - o_t\|_2^2, \tag{8}$$

where $f_{t+1}$ and $f_t$ are the feature maps generated from the new feature extraction network $F_{extract}^{t+1}$ and the old frozen network $F_{extract}^t$, respectively; $(y_{t+1}, y_t)$ and $(o_{t+1}, o_t)$ are the center and offset heatmaps predicted by the object location networks $F_{loc}^{t+1}$ and $F_{loc}^t$ for new and old datasets, respectively; and $(M_f, M_y, M_o)$ refers to the number of activation values in the feature map and prediction outputs.

- The regularization loss, $L_{Reg}$, is adopted in EPicker to avoid overfitting to the old datasets ($D_1 \dots D_t$) in the process of repeatedly minimizing the object detection loss on the exemplar ($E_1 \dots E_t$). The regularization loss term is formulated as follows:

$$L_{Reg} = \sum\|\theta_{t+1} - \theta_t\|_2^2, \tag{9}$$

where $\theta_t$ represents the frozen parameters of the old model $t$, and $\theta_{t+1}$ represents the parameters of the new model $t + 1$. Overtraining on the exemplar is avoided by penalizing significant changes in the parameters between the new and old models, and the general knowledge from the old task is remembered.

**Sparse annotation in EPicker**. Sparse annotation means that only a small number of particles are required to pick for building the training dataset, that is, many positive particles are unlabeled, which can significantly reduce the difficulty and workload of labeling. As in some challenging datasets with extremely small particles, there are often more than 1000 particles on one micrograph, which makes it difficult or impossible to annotate all the positive particles, either by manual picking or 2D classification. However, the unlabeled positive particles can mislead the training process and restrain the detection of good particles, which finally results in the absence of some positive particles. To address the problem of sparse annotation, Topaz[11] proposed the GE-binomial algorithm, which is based on the generalized expectation (GE) criteria. However, Topaz tended to pick a large number of particles, many of which were located on contamination (Supplementary Fig. 6). Some previous works dealing with sparse annotation in natural images solved this problem by reweighting the importance of region proposals generated by the two-stage object detector Faster RCNN[38,39] or recalibrating the loss of the anchors used by the one-stage object detector YOLO[40]. To solve the problem of sparse annotation in EPicker, we need to reduce the penalty on the loss function of these potentially positive particles and try to generate more positive labels to enhance the prediction ability of the detector. On the one hand, we followed the concept of GHM loss[41] and ignored some hard examples that can be seen as outliers. These hard examples are negative samples, however, with high confidence scores in the prediction results of the detector. Ignoring them can reduce the possibility of mistaking unlabeled positive particles as negative particles. On the other hand, we obtained some pseudo labels from the prediction results of the detector with very high confidence scores. Particles with very high confidence scores in the prediction were likely to be unlabeled particles. EPicker automatically generates pseudo labels for these particles and treats them as positive samples. Finally, the final prediction loss of the particle center in the object detection loss of the EPicker is defined as

$$L_k = -\frac{1}{N}\sum_{xy}\begin{cases} (1 - \hat{Y}_{xy})^\alpha \log(\hat{Y}_{xy}) & if\ Y_{xy} = 1\ or\ \hat{Y}_{xy} > \tau_1 \\ (1 - Y_{xy})^\beta (\hat{Y}_{xy})^\alpha \log(1 - \hat{Y}_{xy}) & if\ Y_{xy} = 0\ and\ \hat{Y}_{xy} < \tau_2 \end{cases}, \tag{10}$$

where $\tau_1$ is the threshold for controlling the confidence of the detector to reverse negative particles with high confidence scores into unlabeled positive particles, and $\tau_2$ is the threshold for controlling the confidence of the detector to reduce the substantial penalty on the loss function for the potential unlabeled positive particles, which reduces the detection of particles with high confidence scores as negative samples or background in the training process. EPicker empirically sets $\tau_1 = 0.7$ and $\tau_2 = 0.5$ in the experiments.

**Evaluation of particle picking**. EPicker uses average precision (AP) and average recall (AR) to evaluate the performance of particle picking, which has been widely adopted in the field of object detection[42]. A brief introduction and the settings used in the present work are as follows.

EPicker generates a score for each picked particle to present confidence. When the confidence score was greater than a given threshold, the picked particle was considered correct. Precision and recall were used to evaluate picking. Precision is defined as the ratio of the number of correctly picked particles to the number of all picked particles,

$$precision = \frac{TP}{TP + FP}. \tag{11}$$

Recall is defined as the ratio of the number of correctly picked particles to the number of all correct particles,

$$recall = \frac{TP}{TP + FN}. \qquad (12)$$

Here, true positive (TP) denotes the correctly picked particles, false positive (FP) denotes the incorrectly picked particles, and false negative (FN) denotes the correct particles that are not picked. The intersection over union (IoU) is a measure of the overlapped area between the detected particle and the ground truth,

$$IoU = \frac{area(P_D \cap P_G)}{area(P_D \cup P_G)}, \qquad (13)$$

where $P_D$ represents the detected particles, and $P_G$ represents the ground truth. In all our tests, we used an IoU threshold of 0.5, which means that the detected particles with IoU less than 0.5, are incorrect. When the prediction of particle size was turned off, we assigned a constant particle size for each sample.

Based on the definitions above, precision and recall vary with the setting of the score threshold. To avoid the influence of the threshold setting, AP is calculated as the average precision of all recall values:

$$AP = \int_0^1 P(R)dR, \qquad (14)$$

where precision, P, is considered a function of recall R. The final AP is the average of the AP values for all the testing micrographs. AR is the average of the recall values for all the testing micrographs.

When EPicker is trained in a continual manner, adding more training datasets means merging more features into an old model. The dissimilarity of features between the old and new datasets may influence the effectiveness of merging different features, which is indicated by forgetting some old features. To evaluate this influence, we defined the complexity of a new dataset as how the features in the new dataset match the features in the old datasets. The higher the complexity, the greater the differences. We then tested the relationship between complexity and forgetting rate.

To assess the complexity of a new dataset, we used an old general model to pick new particles. The complexity of the new dataset is inversely proportional to the picking accuracy and is empirically defined as:

$$C = \frac{100}{10^{(AP+AR)}}, \qquad (15)$$

where $C \in [1, 100]$ is the complexity of the new dataset, and AP and AR refer to the average precision and average recall when directly using the old general model to pick new particles. When the features of new particles are significantly different from those in the old datasets, the picking may fail and lead to low AP and AR, and consequently, a high complexity.

The reduction in AP and AR indicates forgetting. We defined the forgetting rate for AP and AR, respectively, as the average reduction in AP and AR of all old datasets before and after training on a new dataset.

**Comparison with other continual learning algorithms**. We compared the continual learning method in EPicker with other widely used continual learning methods.

The method in EPicker was compared with memory-aware synapses (MAS)[21], which is a regularization method that computes the importance of the parameters of a neural network and constrains the changes in important weights. We adapted MAS to EPicker, and the importance weight $\Omega_{ij}$ for parameter $\theta_{ij}$ is defined as:

$$\Omega_{ij} = \frac{1}{N} \sum_{k=1}^{N} \|g_{ij}(x_k)\| = \frac{1}{N} \sum_{k=1}^{N} \left\| \frac{\partial \|F(x_k, \theta)\|_2^2}{\partial \theta_{ij}} \right\|, \qquad (16)$$

where $x$ is the input image, $N$ is the total number of input images, $\theta$ is the parameter of the current model, and $F(x_k, \theta)$ represents the multi-dimensional output function defined in MAS[21]. We evaluated the experimental results by considering the function of the feature extraction network $F_{extract}(\theta)$ and the function of the whole object detection network $F_{OD}(\theta)$ as the output function $F(\theta)$.

Additionally, the method was compared with Faster ILOD[28], which uses Faster R-CNN[34] as the base object detector. Peng et al. used Faster ILOD[28] and proposed an adaptive distillation method to properly train previous knowledge. Because the framework of our network is completely different from that of Faster ILOD, we only adopted the idea of the multi-network adaptive distillation (AD) proposed by Peng et al[28].

As shown in Supplementary Table 3, our method performed better than MAS and Faster ILOD. For MAS, the introduction of important weights led to serious conflicts between the parameters of the old and new models. For Faster ILOD, which dealt with class-incremental learning problems in natural images, the adaptive knowledge distillation method was used on all datasets. Because the particles in a new dataset are often significantly dissimilar to those in old datasets, constraining the new and old models to extract similar features on the new datasets is problematic.

**Fiber picking and tracing algorithm**. EPicker also supports the picking of fibers. For an input micrograph with fibers, EPicker first picks fibers as normal particles, and

then places the coordinates of the picked particles into a point set. We developed a line tracing algorithm (LTA) to link the discrete points of the fibers as lines (Supplementary Fig. 5). The fiber bending angle parameter, *ang*, was used to set an acceptable maximum curvature.

A particle is first randomly selected as the starting point, and then LTA identifies the closest point and connects the two points into a line segment. LTA sets the second point as the starting point, ignores any selected points, and places all the points whose distance from the new starting point is less than a radius, $r$, into a candidate point set. EPicker empirically sets $r = 100$ at a reduced micrograph with a width of 1024 pixels. Then, LTA finds the closest point in the candidate point set and connects it to the new starting point. If the angle between the current line segment and the previous line segment is larger than the angle threshold, the current candidate point is removed from the candidate point set, and the next candidate point is identified and tested. This procedure is repeated until a candidate point satisfying the angle constriction is obtained. If the angle calculated at all candidate points is larger than the angle threshold, a new fiber is initiated. LTA is repeated until no eligible points are in the point set.

The tracing result of the fibers is further smoothed by removing some points on the lines when the angle between the adjacent line segments is less than a certain threshold and is set to 0.1 rad in EPicker.

**Image processing and 2D classification**. The datasets downloaded from EMPIAR were processed using MotionCor2[43] to generate single-frame micrographs, if applicable. The defocus of the micrographs was determined using the CTFFind4[44]. All 2D classifications were performed using the THUNDER[45].

**Reporting summary**. Further information on research design is available in the Nature Research Reporting Summary linked to this article.

## Data availability

Datasets used for the continual learning study are from EMPIAR with entry codes: EMPIAR-10017, EMPIAR-10025, EMPIAR-10028, EMPIAR-10075, EMPIAR-10081, EMPIAR-10089, EMPIAR-10097, EMPIAR-10146, EMPIAR-10203, EMPIAR-10228. The according annotations used for the continual learning is deposited in a public database https://dataverse.harvard.edu/dataverse/EPicker. Dataset used to test biased and unbiased picking are from EMPIAR with entry codes: EMPIAR-10401, EMPIAR-10090. The Fab and liposome dataset is available for downloaded from https://doi.org/10.7910/DVN/I92FFJ and https://doi.org/10.7910/DVN/ZMN57Q, respectively. The remaining datasets used in this study are from EMPIAR with entry codes: EMPIAR-10004, EMPIAR-10033, EMPIAR-10057, EMPIAR-10058, EMPIAR-10093, EMPIAR-10096, EMPIAR-10122, EMPIAR-10168, EMPIAR-10175, EMPIAR-10190, EMPIAR-10192, EMPIAR-10197, EMPIAR-10202, EMPIAR-10205, EMPIAR-10216, EMPIAR-10590, EMPIAR-10406, EMPIAR-10470, EMPIAR-10429, EMPIAR-10241, EMPIAR-10270, EMPIAR-10420, EMPIAR-10407, EMPIAR-10059, EMPIAR-10402, EMPIAR-10399, EMPIAR-10454, EMPIAR-10443, EMPIAR-10379, EMPIAR-10456, EMPIAR-10350, EMPIAR-10335, EMPIAR-10217, EMPIAR-10291, EMPIAR-10290, EMPIAR-10289.

## Code availability

The program code is available for downloading from our project website http://thuem.net and the Github repository https://github.com/thuem/EPicker. Detailed information about software installation and usage can be found at: http://thuem.net.

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

## Acknowledgements

This work was supported by funds from The National Key Research and Development Program (2016YFA0501102 to X.L.), National Natural Science Foundation of China (61871256 to Y.S., and 61673234 to J.C.), Advanced Innovation Center for Structural Biology (to X. L., and Y. S.), Beijing Frontier Research Center for Biological Structure (to X.L.), Tsinghua-Peking Joint Center for Life Sciences (to X. L.). We acknowledge Chao Lin from Tsinghua University for providing the Fab and liposome datasets. We acknowledge Tsinghua University Branch of China National Center for Protein Sciences Beijing for providing facility supports in computation.

## Author contributions

X.L., Y.S., and J.C. initialized the project. X.Z. developed the continual learning algorithm. T.Z. implemented the CenterNet algorithm. X.Z. and T.Z. wrote the program. T.Z. prepared the release package with an integrated Python environment. X.Z., T.Z., and X.L. performed the tests. X.L. developed the graphic user interface. X.L., X.Z., and T.Z. wrote the manuscript. All authors revised the manuscript.

## Competing interests

The authors declare no competing interests.
