## [Peer Review File · Nature Communications]

EPicker is an exemplar-based continual learning approach for knowledge accumulation in cryoEM particle pickingReviewers' Comments:

Reviewer #1:

Remarks to the Author:

In this manuscript, the author created a new pipeline for picking particles from cryo-EM micrographs based on neural networks that can continuously incorporate information from new datasets. Although particle picking is an important step in cryo-EM processing, and this new approach appears to perform as expected. However, I am not sure if the author has demonstrated significant improvements over existing algorithms, nor is the validation comprehensive enough to make a judgement. In my opinion, this work should be published in a more specialized journal.

Comments:

1. Whether computing time of training is the only advantage for this method ? they should compare it with other popular picking methods of similar kinds, for both picking accuracy and training efficiency
2. It is hard to evaluate whether the training set they picked is typical ? whether the advantage of this method still maintains when a larger training set is involved ?
3. How is the exemplar set picked?
4. How is "fine tuning" performed ?
5. Figure 3 and 4 are poorly presented. It's rather difficult to evaluate how well the catastrophic forgetting problem is solved only through example pictures in Fig .3. For Fig. 4, the question is whether the identification of CP1RP by their method is actually accurate ?
6. Recent papers on neural-network based particle picking algorithms such as DRPnet and DeepCryoPicker should be cited and compared.

Reviewer #2:

Remarks to the Author:

The authors propose a deep learning framework for for particle picking in cry-EM. Specifically, the authors show that their proposed exemplar-based continual learning approach performs very similarly to joint training on all datasets, avoiding catastrophic forgetting of old datasets. Further, they show that the fine-tuning approach underperforms and is not able to avoid catastrophic forgetting.

While the proposed method is sound and shows improvement over state-of-the-art solutions, the manuscript shows critical drawbacks.

Foremost, the code is not available at the time of review. the website states a "coming soon" message. This is not acceptable as I was not able to test and validate the proposed method.

In addition, I have some major concerns with the added value of the proposed method, as well as its benchmark:

1) It is not clear what parts of EPicker are drawn from CenterNet, and which ones are novel to EPicker. there are several mentions of the "backbone" of the framework, but sometimes that backbone refers to CenterNet and other times to EPicker. It is very confusing for the reader and one cannot fairly assess the novelty of the proposed method. Also, the schematic figures (fig 1 and 2) could be significantly improved to showcase CenterNet and the novel additions of EPicker. That would make it easier to follow the added value of the manuscript.

2) The manuscript states that EPicker supports Joint, Fine-Tuning and Continual learning. However, the paper demonstrate that Fine-tuning is not desirable as it cannot avoid catastrophic forgetting. Then, why support it in EPicker? Or is it implemented so that users can verify the superior performance of continual learning? In any case, due to the lack of code I could not check this myself.

3) While there is an ablation study on the different losses used in EPicker, there is no mention on how

the hyper-parameters are picked, such as the lambdas, number of layers, etc. This should be clearly stated and analysed. Also, it would be desirable to have these studies performed in more than one combination of datasets. Even more when there seems to be datasets that are more difficult to pick than others (such as E10097).

4) While I am not an expert on Cryo-EM, I would say that mostly using mAP//mAR as a performance metric is not enough. I miss some qualitative assessment of the proposed method beyond the few figures that are shown in the paper.

Finally, it would be good to have a dataset complexity metric as there seems to be datasets that are more difficult to pick than others (such as E10097). A natural question that arises is how does the complexity of the datasets affect the forgetting and continual learning?

Minor comments:

Please expand the figure captions to better describe the figures. Currently they are very short and barely explain what is going on on the figure.

Reviewer #3:

Remarks to the Author:

1-Paper design a deep learning approach for single particle picking in cryo-EM called (EPicker).

2-The deep learning approach is based basically on the continual learning approach to design the EPicker which uses the CenterNet ability of estimating both the position and size of the objects.

3-In addition to the DeepEM [7] Warp [8], Topaz [9],[10], and crYOLO [11], recently there is some other papers in the field used the deep learning approach (fully automated) to solve the particle picking issues in different manner.

4-The design program mainly based on using the an exemplar-based continual learning to extract the key learning features which is going back to the main issues of selecting and using a clean training dataset that is still a big challenge of the deep learning.

5- In DeepCryoPicker, a full automated approach for both training dataset generation based on different automated unsupervised learning is designed which make the program more powerful than depending on a clean training dataset for another learning scheme.

6-Paper is focusing on comparing the results with the other CNNs or deep learning scheme and missing comparing the picking results with the some state-of-the-art approaches in the cryo-em field such as RELION 3.1 and EMAN 2.31.

7-A key feature of the design network is based on using an raining example during the training and is considered as a reference for old features, which preserves the old knowledge and that will be good is the training dataset that been selected be in the same training domain such as shape and size, however, if the training sample differs from dataset to another that will affect the whole training scheme.

8-We need to see some other measurement criteria such as AUC curve, and others such as precision, recall, the accuracy, loss and training plot ect. which is the most important fact to judge the design network.

9-Consuming time for training and testing is missing.

10-We need to see the time consuming for the picking part.

11-is the testing dataset was external which means does the program uses some cryo-EM images for an external data that the deep learning approach did not see during the training.

12-The most important fact in the 3D density map by the single particle picking is the density map resolution, we need to see the FSC curve at the .134 to check the final resolution output the program.

13- You have to compare that as well with the other software's in the filed such as RELION 3.1, and EMAN 2.31.

REVIEWER COMMENTS

Reviewer #1 (Expertise: CryoEM for protein complex characterization):

In this manuscript, the author created a new pipeline for picking particles from cryo-EM micrographs based on neural networks that can continuously incorporate information from new datasets. Although particle picking is an important step in cryo-EM processing, and this new approach appears to perform as expected. However, I am not sure if the author has demonstrated significant improvements over existing algorithms, nor is the validation comprehensive enough to make a judgement. In my opinion, this work should be published in a more specialized journal.

A: The major contribution of this work is to introduce the concept and method of continual learning to current deep-learning-based particle picking methods, which may be also useful for other similar methods of object detection in cryoEM. The objects to be detected in cryoEM are often widely different. While general models (typically trained on 50~100 datasets in public databases) are provided together with the published software packages, the models are often not “general” enough to identify all unseen protein complexes. Accordingly, the results of picking are often not reliable for some challenging datasets, which is also a well-known issue in deep-learning-based methods. Consequently, the template-based methods are still the major methods used in many labs.

We want to change this situation, and attempt to make the deep-learning-based methods as reliable as or even more powerful than the template matching methods in cryoEM. The proposed continual learning method is the solution, which mimics the learning process of the human, and makes the general model more and more powerful during long-term applications. The idea of continual learning is also particularly suitable for the automated cryoEM system in a public cryoEM facility, which processes large number of different data every day.

In summary, the idea of the continual learning is essential to make deep learning play an important role in cryoEM. What’s more, the picking performance of different training modes are rarely discussed in most published particle-picking papers. The present work also made a deep discussion and comparison of different training strategies.

Comments:

1. Whether computing time of training is the only advantage for this method ? they should compare it with other popular picking methods of similar kinds, for both picking accuracy and training efficiency

A: The reduction of training time is not the only advantage of our method. The major advantage of the proposed continual learning strategy is to enhance the performance and reliability of the particle picking. The generalization ability of deep learning method relies on the training on a large number of datasets, that is, without enough features learned from big data, the generalization performance of the deep learning

cannot be guaranteed. The proposed work provides a way to accumulate the knowledge about particle features, avoids catastrophic forgetting on old datasets, and finally improves the picking performance, especially, on unseen datasets. The continual learning has an advantage in computing time, but not the only one. Sorry for the confusion, we have made a major revision to make above points clearer.

More, we have added the comparison with other methods in the revised manuscript. In order to demonstrate the picking performance of EPicker on picking known and unseen particles, we used a sample of Fab (the antigen binding fragment). The Fab is a small protein, ~60 kDa, and we didn't find a similar dataset in the public dataset (so it may not be included in the training datasets of the published software). Determining the structures of small proteins is a very important but challenging task for cryoEM. Due to the small size and high concentration (see the following **Figure R1.1**), more than 2000 Fab particles per micrograph, the particle picking job is challenging. We compared several well-known programs, including template-based RELION, unsupervised-learning-based DRPNet and DeepCryoPicker, and supervised deep-learning-based crYOLO and Topaz (EPicker belongs to this category).

Figure R1.1 A typical micrograph of Fab.

We compared the performance of these programs (please see the following **Figure R1.2** for details) on the Fab dataset. A Ribosome dataset was used to measure the catastrophic forgetting. As expected, without training on the Fab dataset, all programs did not work well. Then, we used EPicker to perform an initial picking, followed by a 2D classification to generate a training dataset and templates. After training on the new Fab training dataset, EPicker, crYOLO, TOPAZ and RELION worked well. Because missing a procedure to import new features, DRPNet and DeepCryoPicker failed in the picking.

Direct picking based on a general model

Picking based on supervised learning model

Unsupervised learning

MODEL

Annotations on training dataset

2D Classification

Template matching

Figure R1.2. Comparison on different software. Because the size of Fab is very small relative to the whole micrograph, in order to improve the visualization, we just show a small region of a raw micrograph. A Fab dataset containing 179 micrographs was used to evaluate different software. A ribosome micrograph was also used. The Fab rarely appears in public databases, and hence is a good “unknown” sample for most published software. In contrast, the ribosome dataset is contained in most of public databases, and thus is a “known” sample for the published software. **a ~ c)** Typical picking results on the ribosome and Fab micrographs of EPicker, crYOLO and TOPAZ, respectively. Here we used general models associated with three software. The general model of EPicker was trained on 50 heterogeneous datasets excluding Fab (please see **Supplementary Table 8** in the revised manuscript). The general model of crYOLO was trained on a combination of 53 datasets. And the number of datasets used to train the general model of TOPAZ is unknown. All these three models show ideal performance on the ribosome, but not ideal on the Fab. **d~e)** Picking results of DRPNNet and DeepCryoPicker on Fab. Both software employed unsupervised-learning algorithms and failed to pick Fabs. No further tests can be performed since a training procedure is not

available. **f**) Ten selected 2D classes and corresponding particle annotations of Fab. To generate a training dataset, the initial picking results of EPicker (shown in **a**) on 8 micrographs in the Fab dataset were filtered by THUNDER 2D classification. In the ten classes, 6562 particles showing obvious features of the Fab were selected as training annotations, and then were used to build the training dataset for EPicker, crYOLO and TOPAZ. The ten class averages were used as the templates for RELION. **g ~ i**) Typical picking results on the ribosome and Fab micrographs of EPicker, crYOLO and TOPAZ, respectively, after loading the corresponding general models and training on the newly built Fab dataset. The continual learning was used for EPicker and the fine-tuning was used for crYOLO and TOPAZ. All three software performed as expected on the Fab. For crYOLO and TOPAZ, the performance on the ribosome dataset was not maintained, indicated by lots of missed picking on the ribosome. While the picking results of EPicker were similar to those before the training on Fab. **j**) A typical picking result of RELION based on template matching using the ten class averages. RELION cannot avoid picking ice particles and hence did not perform as well as the other three deep-learning-based software. **k ~ n**) The results of 2D classification on the picking results of EPicker, crYOLO, TOPAZ and RELION, corresponding to the processing in **g ~ j**. All software picked ~ 600,000 particles (details shown on the bottom of each figure). After empirically selecting classes well centered and with secondary-structure details, EPicker and RELION output more good particles (~390,000) than the other two software.

In the tests, while the Fab (~60 kDa) and the ribosome (~3MDa) have very different shapes and sizes, EPicker can maintain the performance on the ribosome dataset after performing continual learning process on the Fab dataset. crYOLO and TOPAZ lost their ability to pick ribosome after fine-tuning on Fab. RELION cannot work without an initial picking to generate the templates. We used the general model of EPicker to generate the required templates for RELION. This also implies that the deep-learning-based method can reduce more labors than the traditional methods. We failed to pick Fab particles using unsupervised learning methods DRPNet and DeepCryoPicker.

We also further compared the final picking performance of EPicker, crYOLO, TOPAZ and RELION, and the corresponding results are shown in **Figure R1.2 g, h, i** and **j**, respectively. All of EPicker, crYOLO, TOPAZ and RELION picked 560,000 ~ 600,000 particles on 179 micrographs. The following 2D classifications with 100 classes show similar results (see **Figure R1.2k ~ n**). Because we don't know the ground truth of the Fab, we used the number of particles in the classes showing features of the secondary structures as our metric (empirically selected by visual inspection). EPicker and RELION showed slightly better results than crYOLO and TOPAZ. Compared with RELION and TOPAZ, EPicker and crYOLO can effectively avoid picking ice particles (please see the ice particles in **Figure R1.2g ~ j**). This semi-quantitative comparison demonstrates that EPicker has advantages in both picking accuracy and training efficiency.

We also compared the training time and picking time of different software (see the following **Table R1.1**). EPicker isn't slower than other programs.

All these new evaluation results have been added into the revised manuscript.

Table R1.1 Comparison of training time and picking time of different software. To evaluate the training process, we compared different software on a combination of 10 datasets (used in **Table 1** of the submitted manuscript), including 100 micrographs and 19300 annotated particles. All the experiments were performed on a single GPU

(RTX 2080Ti). The time of joint training on 10 datasets was measured for crYOLO, Topaz and EPicker. For EPicker, the time for continual learning was measured by adding 10th dataset to a general model trained on the previous 9 datasets. RELION wasn't included in the comparison. To evaluate the picking process, the average time for picking a single micrograph was recorded.

Software	Training time (100 micrographs)	Picking time (1 micrograph)
TOPAZ (Joint Training)	30 min	0.5 s
crYOLO (Joint Training)	56 min	0.3 s
DRPNet	×	3 s
DeepCryoPicker	×	30 s
EPicker (Joint Training)	50 min	0.3 s
EPicker (Continual Learning)	20 min	0.3 s

2、 It is hard to evaluate whether the training set they picked is typical ?

A: In our experiments, we chose 10 different datasets from EMPIAR, including soluble proteins, ion channels, ribosome, and virus, size ranging from 100kDa to several MDa, as shown in the following **Table R1.2**. For the training dataset, we tried to include particles as diverse as possible (a list of sample names is shown in **Supplementary Table 8** of the revised manuscript), considering size, shape, and structural features.

Table R1.2. Illustration of images in different training datasets after local enlargement.

E10089 (TcdA1)	E10146 (Apo ferritin)	E10028 (80S ribosome)	E10203 (Nodavirus)	E10025 (20S Proteasome)
				E10017 (Beta-galactosidase)	E10097 (Influenza hemagglutinin trimer)	E10075 (Phage MS2)	E10081 (HCN1 ion channel)	E10228 (Phosphodiesterase 6)
				
whether the advantage of this method still maintains when a larger training set is involved?

A: Our continual learning test for Fab on a general model trained on 50 datasets excluding Fab (**Supplementary Table 8** of the revised manuscript) demonstrated the advantage of the proposed method. The yielded general model can be further trained on more datasets. We made a semi-quantitative analysis to give a deeper insight into the behavior of the continual learning.

Adding more training datasets means merging more features into an old model trained on the combination of old datasets. The dissimilarity of features between the old and new datasets may influence the effectiveness of merging different features (knowledge), which is indicated by forgetting some old features. To evaluate this influence, we defined the complexity of a new dataset as how the features in the new dataset match the features in the old datasets. The higher the complexity, the greater the differences. We then tested the relationship between complexity and forgetting rate.

The complexity and the forgetting rate metrics are formulated as follows:

- **The complexity.**

To assess the complexity of a new dataset, we use an old general model to pick new particles. The complexity of the new dataset is inversely proportional to the picking accuracy, and is empirically defined as

$$C = \frac{100}{10^{(AP+AR)}}$$

where $C \in [1,100]$ is the complexity of the new dataset, AP and AR refer to the average precision and average recall when directly using the old general model to pick new particles. When the features of new particles are significantly different from those in the old datasets, the picking may fail and lead to low AP and AR, and consequently, a high complexity.

- **The forgetting rate.**

The reduction in AP and AR indicates forgetting. We defined the forgetting rate for AP and AR, respectively, as the average reduction in AP and AR on all old datasets before and after training on a new dataset.

For the evaluation, we still used the ten datasets (used in **Table 1** of the submitted manuscript). The AP/AR values used to calculate the complexity and the forgetting rate are shown in the following **Table R1.3**.

Table R1.3. Evaluation of particle picking (AP/AR) under different training modes. The 10 datasets are indicated by D1 ~ D10, the corresponding exemplar dataset are indicated by E1 ~ E10. The joint training (JT) and continual learning (CL) are tested with different datasets. The cells with grey background indicate that the corresponding dataset is unseen for the old model. The cells with underline values indicate that the corresponding dataset is newly added for the model.

Name	E10089	E10146	E10028	E10203	E10025	E10017	E10097	E10075	E10081	E10228
NO.	(1)	(2)	(3)	(4)	(5)	(6)	(7)	(8)	(9)	(10)
JT(D ₁ ~D ₅)	97.5/99.1	96.4/97.2	96.7/97.5	93.1/98.9	92.0/96.6	19.8/50.2	9.7/64.9	95.1/98.7	80.9/92.3	26.5/77.6
CL	96.4/99.2	94.9/97.2	96.4/97.7	90.2/98.3	90.3/95.5	96.2/98.5	66.5/97.2	92.2/96.7	84.4/95.2	55.9/97.0
(E ₁ ~E _t +	96.1/98.4	96.2/97.6	96.6/97.3	89.4/97.7	90.2/95.7	96.4/98.8	93.9/98.5	92.2/96.4	88.6/96.9	84.1/97.4
D _{t+1})	96.1/98.8	96.7/97.9	96.6/97.3	92.6/98.9	89.9/95.3	96.4/99.0	91.9/96.8	95.7/96.7	80.1/95.0	83.4/96.4
t = 5 ~ 9.	96.4/98.8	96.7/97.7	96.7/97.3	91.9/98.9	89.7/95.8	95.8/98.3	92.8/97.5	96.4/97.1	94.9/97.7	80.8/96.8
	96.4/98.8	96.8/97.6	96.6/97.3	90.1/98.9	89.6/95.6	94.6/97.9	91.8/97.5	95.8/96.7	93.1/96.9	91.8/97.2

A plot is calculated based on the data in **Table R1.3**, and shown in the following **Figure R1.3**.

Figure R1.3. Relation between complexity of new dataset and forgetting rate on the old datasets. The complexity (yellow) and the forgetting rate of AP (blue) and AR (light blue) for sequentially added new dataset from D6 to D10.

It can be seen from **Figure R1.3** that the complexity has strong correlation with the forgetting rate. Dataset 6 (Beta-galactosidase in **Table R1.2**) has the highest complexity, which does have very different features from dataset 1~5 (see the first row in **Table R1.2**). While the high complexity of dataset 6 causes a forgetting rate up to 1.5%, the forgetting is still minor comparing with the absolute AP value of $\sim 90\%$. The low complexities of dataset 7~10 just cause a subtle forgetting, and even improve the performance of picking on the old datasets (Dataset 7~9, indicated by the negative forgetting rates).

In summary, adding new datasets with different features (complexity) won't cause significant forgetting. Meantime, adding datasets with low complexity sometimes can improve the performance on the model. Therefore, with larger and larger training datasets involved, EPicker can maintain the performance of the model on the old datasets.

3、 How is the exemplar set picked?

A: The exemplar set is randomly selected from the training dataset. In practice, the user usually chooses some high-quality particles and micrographs for training. The quality of the training dataset is easily guaranteed.

Random selection is enough for EPicker. We tested the influence of different selection strategies of the exemplar dataset, including the selection of micrographs with the highest picking accuracy and random selection. The exemplar dataset was first generated during training a model on dataset 1 ~ 5. Then, we used dataset 6 as a new dataset with the exemplar to continually trained the old model. Finally, we measured the picking performance (AP/AR) on all involved datasets (see **Table R1.4**). It can be seen from the table that the selection strategies won't cause obvious difference, where strategy 1 indicates selecting samples with the highest picking accuracy, and the other three strategies are random selection.

Table R1.4. Comparison of different selection strategies for constructing an exemplar set. An initial model was

trained on micrographs from five EMPIAR datasets: 1–5. Then, EMPIAR-10017 was used for continual learning based on the initial model. AP/AR values of the picking are shown in the table.

Strategy	E10089	E10146	E10028	E10203	E10025	E10017	mAP/mAR
	D1	D2	D3	D4	D5	D6	
1	96.2/99.1	95.1/97.3	96.6/97.7	91.1/98.0	90.3/96.0	96.0/98.1	94.2/97.8
2	96.3/99.0	94.5/97.2	96.8/97.2	90.0/98.5	90.1/95.5	96.2/98.5	94.0/97.7
3	95.2/98.9	95.1/97.4	96.6/97.5	89.9/98.0	90.3/95.8	96.1/98.0	93.9/97.6
4	96.4/99.2	94.9/97.2	96.4/97.7	90.2/98.3	90.3/95.5	96.2/98.5	94.1/97.7

4、 How is “fine tuning” performed ?

A: Compared with the dual-path network used for the continual learning, the network used for fine-tuning uses a single-path network. The single-path network has the same structure as one branch of the continual learning dual-path network, and lacks a reference network for maintaining the old knowledge. During the fine-tuning process, the parameters of all network layers are modified to ensure better performance on the new datasets. After the training, the fine-tuning model cannot extract the old particle features.

We have made a detailed discussion of fine-tuning in a new section “Catastrophic forgetting in fine-tuning” of the revised manuscript.

5、 Figure 3 and 4 are poorly presented. It’s rather difficult to evaluate how well the catastrophic forgetting problem is solved only through example pictures in Fig .3.

A: Sorry for the confusing. **Figure 3** is used to demonstrate the catastrophic forgetting problem. **Figure 4** is used to compare the characteristics of different training modes without considering the catastrophic forgetting. By comparing the differences between continual learning and fine-tuning, EPicker provides more choices to find interested particles. We have revised the manuscript to make these points clearer.

For **Figure 3**, we modified the labels, and used different colors of boxes to label the particles. In **Fig. 3b, c and d**, the red, blue and yellow boxes indicate the particles were correctly picked, wrongly picked and missed in picking, respectively. The revised **Figure 3** is as follow.

And we also rewrote the corresponding paragraph, and renamed the title as “Catastrophic forgetting in fine-tuning” to discuss fine-tuning and catastrophic forgetting in more detail.

Figure 3 Comparison of different particle picking methods and illustration of catastrophic forgetting. The picked particles on a micrograph of 80S ribosome (EMPIAR-10028) are annotated by square boxes. The red, blue and yellow boxes indicate the particles were correctly picked, wrongly picked and missed in picking, respectively. **a)** The manually picked particles were considered as the ground truth. **b)** The joint-training model resulted in picking 97 % of the ground-truth particles. **c)** The continual-learning model led to similar result as that of using the joint-training model; 96 % of the ground-truth particles were picked; **d)** The picking with the fine-tuning model missed 35 % of the ground-truth particles, indicating catastrophic forgetting.

For Fig. 4, the question is whether the identification of CP1RP by their method is actually accurate?

A: The identification of CP1RP and other particles is accurate in statistics, that is, the relative ratios of the numbers of particles are statistically accurate, but not accurate in actually number of particles. The dataset used in the experiment has very low contrast and contains many bad particles and contaminations, and thus it is hard to accurately obtain the ground truth by manually picking. Therefore, we used 2D classification (calculated by THUNDER) to count the number of particles. The 2D classification in THUNDER is based on K-means regulated by Bayesian inference, which cannot guarantee that all particles classified into a class actually belong to this class, but can guarantee most particles are correctly classified. Empirically, the ratio of the number of particles in the dataset can be reliably determined. Meanwhile, CP2RP, CP1RP and CP particles have significantly different shapes and sizes, and hence can be easily classified with less errors.

6. Recent papers on neural-network based particle picking algorithms such as DRPnet and DeepCryoPicker should be cited and compared.

A: Thanks for the suggestion. We have cited these software and added the comparison to the revised manuscript, please see the beginning part of the response.

Reviewer #2 (Expertise: Deep learning, transfer learning, biological data):

The authors propose a deep learning framework for for particle picking in cry-EM. Specifically, the authors show that their proposed exemplar-based continual learning approach performs very similarly to joint training on all datasets, avoiding catastrophic forgetting of old datasets. Further, they show that the fine-tuning approach underperforms and is not able to avoid catastrophic forgetting.

While the proposed method is sound and shows improvement over state-of-the-art solutions, the manuscript shows critical drawbacks.

Foremost, the code is not available at the time of review. the website states a "coming soon" message. This is not acceptable as I was not able to test and validate the proposed method.

A: Sorry for the inconvenience in accessing the code. We have submitted the manuscript with the downloading address. The website is now available through <http://www.thuem.net/software/epicker/overview.html>. The code and guidance documents are provided together with a tutorial.

In addition, I have some major concerns with the added value of the proposed method, as well as its benchmark:

1) It is not clear what parts of EPicker are drawn from CenterNet, and which ones are novel to EPicker. there are several mentions of the "backbone" of the framework, but sometimes that backbone refers to CenterNet and other times to EPicker. It is very confusing for the reader and one cannot fairly assess the novelty of the proposed method. Also, the schematic figures (fig 1 and 2) could be significantly improved to showcase CenterNet and the novel additions of EPicker. That would make it easier to follow the added value of the manuscript.

A: Sorry for the confusion, and thanks for pointing out the problems. We have revised the manuscript to make the description clearer.

The novel part in EPicker is the design of the dual-path network, the use of the exemplar and corresponding loss function for the continual learning. The dual-path network is composed of two branches A and B with the same network structure. Each of the branches adopts the structure of the CenterNet detector. CenterNet does not support continual learning, while the dual-path network in EPicker enables continual learning. One branch in the network is used for maintaining the old knowledge in the old model, and another one is for generating the new model. While we used CenterNet in our work, the dual-path network does not depend on the CenterNet detector, that is, other similar object detection network can also be used in the dual-path design. We have made a major revision to make these novel points clearer.

The "backbone" in EPicker refers to the feature extraction sub-network, which is one of the two CenterNet components (a feature extraction sub-network and an object location sub-network). Now we have removed the word "backbone" and directly used the terminology "the feature extraction sub-network" in the revised manuscript.

We have redrawn **Figure 1** and **Figure 2** in the revised manuscript to show more details about the network. **Figure 1** focuses on the dual-path neural network. **Figure 2** illustrates the architecture of CenterNet detector used in EPicker.

Figure 1. Architecture of EPicker and schematic of continual learning.

Figure 2. Architecture of CenterNet detector in EPicker.

2) The manuscript states that EPicker supports Joint, Fine-Tuning and Continual learning. However, the paper demonstrate that Fine-tuning is not desirable as it cannot avoid catastrophic forgetting. Then, why support it in EPicker? Or is it implemented so that users can verify the superior performance of continual learning? In any case, due to the lack of code I could not check this myself.

A: Sorry, we did not address the purpose clearly. We now made a major revision for the section of “Biased and unbiased picking” to clarify why fine-tuning is also useful. Continual learning aims to create a general model, which is expected to pick everything like a “particle”. The fine-tuning model is specific to given objects, rather than the general objects. That is, the fine-tuning is necessary in the case that the user does not

want to pick all particles, and just wants to pick part of the particles with given features. Therefore, by supporting the fine-tuning together with the continual learning, EPicker provides a complete tool for particle picking.

3) While there is an ablation study on the different losses used in EPicker, there is no mention on how the hyper-parameters are picked, such as the lambdas, number of layers, etc. This should be clearly stated and analysed.

A: Thanks for the suggestion. We have added more explanations and experiments for the setting of hyper-parameters and the number of layers in the revised manuscript. Some details and evaluation results are following.

The loss function of EPicker is defined as

$$L_{Total} = L_k + \lambda_{off} * L_{off} + \lambda_{size} * L_{size} + \lambda_d * L_{Distill} + \lambda_r * L_{Reg}$$

which consists of four hyper-parameters. The first two hyper-parameters control the weight of prediction precision of particle size and local offset. In the original paper of CenterNet, the authors analyzed the influence of the two hyper-parameters and set $\lambda_{off} = 1$, $\lambda_{size} = 0.1$ (Zhou, X., Wang, D. & Krhenbühl, P. Objects as Points (2019)). We just followed their settings. The latter two hyper-parameters are weighting factors that control the balance between continual learning and location prediction accuracy of particles. When the two weighting factors are large, the network penalizes the change of parameters in the model, which greatly avoids catastrophic forgetting. However, the picking accuracy on the new datasets will decrease (see the first row of **Table R2.1**). On the contrary, when the weighting factors are small, the gradient descent direction of the network will tend to improve the performance on the new datasets, resulting in a decline in the performance on old datasets (see the third row of **Table R2.1**). Therefore, we empirically chose the parameter set $\lambda_d = 0.1$ and $\lambda_r = 0.01$ in EPicker to balance the above problems (see the second row of **Table R2.1**).

Table R2.1. Comparison of different weighting factors of continual learning. An initial model was trained on micrographs from five EMPIAR datasets: 1–5. Then, EMPIAR-10017 was used for continual learning based on the initial model. EPicker chose the weighting factors $\lambda_d = 0.1$ and $\lambda_r = 0.01$, which obtained the highest mAP and mAR.

Strategy	E10089 (1)	E10146 (2)	E10028 (3)	E10203 (4)	E10025 (5)	E10017 (6)	mAP/mAR
$\lambda_d = 1, \lambda_r = 0.1$	96.8/99.2	95.0/97.5	96.8/98.0	90.8/98.3	90.6/95.5	94.2/97.1	94.0/97.6
$\lambda_d = 0.1, \lambda_r = 0.01$	96.4/99.2	94.9/97.2	96.4/97.7	90.2/98.3	90.3/95.5	96.2/98.5	94.1/97.7
$\lambda_d = 0.01, \lambda_r = 0.001$	95.5/98.8	94.2/96.8	96.1/97.3	89.2/97.0	90.0/95.1	96.8/99.0	93.6/97.3

We also compared the feature extraction sub-networks with different number of layers (**See Table R2.2**). For deep learning, increasing the number of network layers means increasing the capacity and learning ability of the network. However, more layers do not always result in better performance, and may lead to a burden of storage space, high computational cost and over fitting problem. As shown in the experimental results (**See Table R2.2**), compared with ResNet of the same number of layers, DLA improves the ability of feature representation by adding more skip connections. A DLA

network with 34 layers (DLA-34) exhibits the best performance for the particle picking task. Hence, EPicker choses DLA-34 as the feature extraction network.

Table R2.2. Comparison of different feature extraction networks. The values of AP/AR based on the models of joint training on six datasets. Two types of feature extraction networks, ResNet and DLA, were evaluated. The numbers following the network names denote the number of layers used in the corresponding neural network.

Network	E10089 (1)	E10146 (2)	E10028 (3)	E10203 (4)	E10025 (5)	E10017 (6)	mAP/mAR
ResNet-18	97.6/99.1	97.0/97.6	96.8/97.7	94.1/97.7	89.3/94.6	92.6/95.8	94.6/97.1
ResNet-34	97.1/98.9	96.7/97.5	96.7/97.9	94.9/98.9	91.5/95.9	96.3/98.7	95.5/98.0
DLA-18	98.1/99.4	96.6/97.3	96.9/98.0	95.0/98.9	92.2/96.7	96.1/99.3	95.8/98.3
DLA-34	97.5/99.1	96.8/97.5	97.6/98.0	95.3/99.4	92.0/96.7	97.0/99.5	96.0/98.4
DLA-46	97.6/99.2	95.9/96.8	97.6/98.6	94.9/99.4	91.8/96.1	95.2/99.3	95.5/98.2

Also, it would be desirable to have these study performed in more than one combination of datasets. Even more when there seems to be datasets that are more difficult to pick than others (such as E10097).

A: Our understanding for this question is that multiple datasets are added for the continual learning in different orders. In **Table 1** of the submitted manuscript, an initial model was first trained on micrographs from five EMPIAR datasets: E10089, E10146, E10028, E10203, and E10025. Then, five new datasets, E10017, E10097, E10075, E10081, and E10228 were used for further continual learning one by one. (The following is a copy of the **Table 1**)

Table 1. Evaluation of particle picking (AP/AR). The 10 datasets are indicated by D_1 – D_{10} , the corresponding exemplar dataset are indicated by E_1 – E_{10} . The joint training (JT) and continual learning (CL) are tested with different datasets. The cells with grey background indicate that the corresponding dataset is unseen for the model. The cells with underlined values indicate that the corresponding dataset is newly added for the model.

Name	E10089	E10146	E10028	E10203	E10025	E10017	E10097	E10075	E10081	E10228	$mAP_{0.5}$
NO.	(1)	(2)	(3)	(4)	(5)	(6)	(7)	(8)	(9)	(10)	$/mAR_{0.5}$
JT(D_1 – D_{10})	97.1/99.1	96.3/97.2	96.8/97.5	93.5/98.3	92.1/96.7	96.8/99.4	92.7/97.2	96.0/96.4	95.3/97.6	92.5/97.4	94.9/97.7
JT(D_1 – D_3)	97.5/99.1	96.4/97.2	96.7/97.5	93.1/98.9	92.0/96.6	19.8/50.2	9.7/64.9	95.1/98.7	80.9/92.3	26.5/77.6	95.1/97.9
CL	96.4/99.2	94.9/97.2	96.4/97.7	90.2/98.3	90.3/95.5	96.2/98.5	66.5/97.2	92.2/96.7	84.4/95.2	55.9/97.0	94.1/97.7
(E_i – $E_i + D_{i+1}$)	96.1/98.4	96.2/97.6	96.6/97.3	89.4/97.7	90.2/95.7	96.4/98.8	93.9/98.5	92.2/96.4	88.6/96.9	84.1/97.4	94.1/97.7
$t = 5-9$.	96.1/98.8	96.7/97.9	96.6/97.3	92.6/98.9	89.9/95.3	96.4/99.0	91.9/96.8	95.7/96.7	80.1/95.0	83.4/96.4	94.5/97.6
	96.4/98.8	96.7/97.7	96.7/97.3	91.9/98.9	89.7/95.8	95.8/98.3	92.8/97.5	96.4/97.1	94.9/97.7	80.8/96.8	94.6/97.7
	96.4/98.8	96.8/97.6	96.6/97.3	90.1/98.9	89.6/95.6	94.6/97.9	91.8/97.5	95.8/96.7	93.1/96.9	91.8/97.2	93.7/97.4

To test the influence of adding dataset in different orders on continual learning, we changed the order of the involved datasets (see **Table R2.3**), i.e., first trained an initial model on micrographs from five EMPIAR datasets: E10097, E10089, E10203, E10025, and E10081. Then, five new datasets, E10146, E10228, E10017, E10075, and E10028 were used for continual learning. Compared with the corresponding values in Table 1, the new results are nearly identical, which demonstrated that the different orders will

not affect the results of continual learning in EPicker.

Table R2.3. Evaluation of particle picking (AP/AR). The 10 datasets are indicated by D1 ~ D10, the corresponding exemplar dataset is indicated by E1 ~ E10. The joint training (JT) and continual learning (CL) are tested with different datasets. The cells with grey background indicate that the corresponding dataset is unseen for the old model. The cells with underline values indicate that the corresponding dataset is newly added for the model.

Name	E10097	E10089	E10203	E10025	E10081	E10146	E10228	E10017	E10075	E10028	$mAP_{0.5}$
NO.	(1)	(2)	(3)	(4)	(5)	(6)	(7)	(8)	(9)	(10)	$/mAR_{0.5}$
JT(D ₁ -D ₁₀)	92.7/97.2	97.1/99.1	93.5/98.3	92.1/96.7	95.3/97.6	96.3/97.2	92.5/97.4	96.8/99.4	96.0/96.4	96.8/97.5	94.9/97.7
JT(D ₁ -D ₅)	93.0/97.8	97.1/98.9	93.3/98.9	92.3/96.7	95.5/97.6	75.9/88.2	86.5/97.9	82.5/92.1	91.6/95.1	94.0/96.5	92.4/81.7
CL	90.0/97.0	96.2/98.8	89.5/98.5	91.6/96.5	94.9/97.4	95.8/96.8	70.6/89.2	88.7/97.8	94.2/96.1	93.8/95.7	93.0/97.5
(E ₁ -E ₁ +	90.6/96.3	96.5/97.2	92.2/98.9	91.5/96.5	93.3/97.1	96.5/97.2	92.9/98.5	69.3/86.4	93.6/96.7	93.8/95.7	93.4/97.4
D ₁ +)	93.7/98.3	95.1/98.8	89.4/98.9	91.6/99.6	93.2/97.3	94.8/96.8	92.6/98.3	96.5/98.9	92.1/97.1	91.3/95.9	93.4/98.4
t = 5 ~ 9.	91.5/97.3	95.3/98.0	91.6/98.9	91.1/96.3	93.1/98.1	96.1/97.6	91.2/97.6	97.2/99.2	96.8/97.7	92.2/96.5	93.8/97.9
	91.1/97.7	96.8/98.6	91.8/98.3	91.3/96.4	92.2/97.1	96.1/97.5	90.8/97.6	97.0/99.0	95.9/96.7	97.6/98.0	94.1/97.7

4) While I am not an expert on Cryo-EM, I would say that mostly using mAP//mAR as a performance metric is not enough. I miss some qualitative assessment of the proposed method beyond the few figures that are shown in the paper.

A: In addition to the mAP/mAR, we also evaluated the variation curve of training loss (See **Figure R2.1**), and calculated the Area Under Curve (AUC) and F1-score as performance metrics (see **Table R2.4**). AUC is defined as the area enclosed by the coordinate axis under the ROC curve. F1-score is the harmonic average of precision value and recall value. All these metrics show the similar tendency as mAP/mAR. The more accurate the picking is, the higher these metrics are. Hence, the mAR/mAP is sufficient to reflect the particle picking performance of EPicker. Therefore, we did not include these metrics in the manuscript. Details are shown in **Figure R2.1**.

Figure R2.1. Metrics to evaluate the continually learning on 5 datasets. We first loaded an initial model, which was trained on micrographs from five EMPIAR datasets: 1–5. Then, EMPIAR-10017 was used for continual learning based on the initial model. **a)** Variation curve of training loss. With the increase of training time, the training loss decreased gradually and finally tended to converge. **b)** Variation curves of AP and AR on the training dataset. With the increase of training time, AP and AR values on the training dataset increased and finally converged. **c)** Variation curves of AP and AR on the test dataset. With the increase of training time, AP and AR values on the test dataset increased and finally converged. **d)** Precision-Recall (PR) curve on the test dataset of EMPIAR-10017. **e)** Receiver Operating Characteristic (ROC) curve on the test dataset of EMPIAR-10017.

Table R2.4. Evaluation results of the continual learning model in EPicker. We first loaded an initial model, which was trained on micrographs from five EMPIAR datasets: 1–5. Then, EMPIAR-10017 was used for continual learning based on the initial model. The AP, AR, F1-score and AUC for all the test datasets are recorded.

Dataset	AP	AR	F1-score	AUC
E10089	96.4	99.2	97.8	97.4
E10146	94.9	97.2	96.0	95.5
E10028	96.4	97.7	97.0	97.3
E10203	90.2	98.3	94.1	96.3
E10025	90.3	95.5	92.8	93.5
E10017	96.2	98.5	97.3	96.3

Finally, it would be good to have a dataset complexity metric as there seems to be datasets that are more difficult to pick than others (such as E10097). A natural question that arise is how does the complexity of the datasets affect the forgetting and continual learning?

A: Thanks for the suggestion. Using a complexity metric is a very good idea. In the revised manuscript, we defined the complexity of a dataset and the forgetting rate as follows.

The dissimilarity of features between the old and new datasets may influence the effectiveness of merging different features (knowledge), which is indicated by forgetting some old features. To evaluate this influence, we defined the complexity of a new dataset as how the features in the new dataset match the features in the old datasets. The higher the complexity, the greater the differences. We then tested the relationship between complexity and forgetting rate using 10 datasets (**Table R2.5**).

Table R2.5. Illustration of images in different training datasets after local enlargement.

E10089 (TedA1)	E10146 (Apoferitin)	E10028 (80S ribosome)	E10203 (Nodavirus)	E10025 (20S Proteasome)
				E10017 (Beta-galactosidase)	E10097 (Influenza)	E10075 (Phage MS2)	E10081 (HCN1 ion channel)	E10228 (Phosphodiesterase 6)

The complexity and the forgetting rate metrics are formulated as follows:

- **The complexity.**

To assess the complexity of a new dataset, we use an old general model to pick new particles. The complexity of the new dataset is inversely proportional to the picking accuracy, and is empirically defined as

$$C = \frac{100}{10^{(AP+AR)}}$$

where $C \in [1,100]$ is the complexity of the new dataset, AP and AR refer to the average precision and average recall when directly using the old general model to pick new particles. When the features of new particles are significantly different from those in the old datasets, the picking may fail and lead to low AP and AR, consequently, the high complexity.

- **The forgetting rate.**

The reduction in AP and AR indicates forgetting. We defined the forgetting rate for AP and AR, respectively, as the average reduction in AP and AR of all old datasets before and after training on a new dataset.

For the evaluation, we still used the ten datasets (**Table R2.5**). The AP/AR values used for calculating the complexity and the forgetting rate are shown in the following **Table R2.6**.

Table R2.6. Evaluation of particle picking (AP/AR) under different training modes. The 10 datasets are indicated by D1 ~ D10, the corresponding exemplar dataset are indicated by E1 ~ E10. The joint training (JT) and continual learning (CL) are tested with different datasets. The cells with grey background indicate that the corresponding dataset is unseen for the old model. The cells with underline values indicate that the corresponding dataset is newly added for the model.

Name	E10089	E10146	E10028	E10203	E10025	E10017	E10097	E10075	E10081	E10228
NO.	(1)	(2)	(3)	(4)	(5)	(6)	(7)	(8)	(9)	(10)
JT(D ₁ ~D ₅)	97.5/99.1	96.4/97.2	96.7/97.5	93.1/98.9	92.0/96.6	19.8/50.2	9.7/64.9	95.1/98.7	80.9/92.3	26.5/77.6
CL	96.4/99.2	94.9/97.2	96.4/97.7	90.2/98.3	90.3/95.5	96.2/98.5	66.5/97.2	92.2/96.7	84.4/95.2	55.9/97.0
(E ₁ ~E _i +	96.1/98.4	96.2/97.6	96.6/97.3	89.4/97.7	90.2/95.7	96.4/98.8	93.9/98.5	92.2/96.4	88.6/96.9	84.1/97.4
D _{i+1})	96.1/98.8	96.7/97.9	96.6/97.3	92.6/98.9	89.9/95.3	96.4/99.0	91.9/96.8	95.7/96.7	80.1/95.0	83.4/96.4
t = 5 ~ 9.	96.4/98.8	96.7/97.7	96.7/97.3	91.9/98.9	89.7/95.8	95.8/98.3	92.8/97.5	96.4/97.1	94.9/97.7	80.8/96.8
	96.4/98.8	96.8/97.6	96.6/97.3	90.1/98.9	89.6/95.6	94.6/97.9	91.8/97.5	95.8/96.7	93.1/96.9	91.8/97.2

A plot is calculated based on the data in **Table R2.6**, and shown in the following **Figure R2.2**.

Figure R2.2. Relation between the complexity of new dataset and the forgetting rate on the old datasets. The complexity (yellow) and the forgetting rate of AP (blue) and AR (light blue) for sequentially added new dataset from D6 to D10.

It can be seen from **Figure R2.2** that the complexity has strong correlation with the forgetting rate. Dataset 6 (Beta-galactosidase in **Table R2.5**) has the highest complexity, which does have very different features from those of dataset 1 ~ 5 (see the first row in **Table R2.6**). While the high complexity of dataset 6 causes a forgetting rate up to 1.5%, the forgetting is still minor comparing with the absolute AP value of ~90%. The low complexities of dataset 7 ~ 10 just cause subtle forgetting, and even improve the performance of picking (Dataset 7 ~ 9, indicated by the negative forgetting rates).

In summary, adding new dataset with different features (complexity) won't cause significant forgetting. Meantime, adding datasets with low complexity sometimes can improve the performance on the model. Therefore, with larger and larger training datasets involved, EPicker can maintain the performance of the model on the old datasets.

Minor comments:

Please expand the figure captions to better describe the figures. Currently they are very short and barely explain what is going on on the figure.

A: Thank you for your suggestion. We have expanded all the figure captions and explained the meaning of each figure in detail.

Reviewer #3 (Expertise: deep learning for the analysis of cryoEM data):

1-Paper design a deep learning approach for single particle picking in cryo-EM called (EPicker).

A: This is true.

2-The deep learning approach is based basically on the continual learning approach to design the Epicke which uses the CenterNet ability of estimating both the position and size of the objects.

A: This is true.

3-In additional to the DeepEM [7] Warp [8], Topaz [9],[10], and crYOLO [11], recently there is some other papers in the filed used the deep learning approach (fully automated) to solve the particle picking issues in different manner.

A: This is true. Compared with these software, EPicker is highlighted by its continual learning ability and multiple picking modes for particles, vesicles and fibers. These new features made EPicker a more reliable, accurate and convenient tool.

4-The design program mainly based on using the an exemplar-based continual learning to extract the key learning features which is going back to the main issues of selecting and using a clean training dataset that is still a big challenge of the deep learning.

A: EPicker is just designed to solve the difficulties and challenges in obtaining the training dataset.

Several related key features of EPicker in order to solve these problems are as follows. 1) EPicker supports sparse annotations by a specially designed loss function, i.e., just a small part of particles on micrographs are needed to be annotated for training. Moreover, 5~10 micrographs are usually enough for training. Generating a training dataset by manually picking a small number of particles or by using the result of a 2D classification method are supported. 2) EPicker only requires positive annotations, and no longer needs negative annotations any more. 3) EPicker supports different training modes and hence allows biased and unbiased picking. The users can pick specific or general particles according to their needs. 4) The continual learning disperses the training on a large number of datasets into multiple training processes and thus it is not necessary to finish the training at one time any more.

Combining the features above, the training process in EPicker becomes efficient and simple.

5- In DeepCryoPicker, a full automated approach for both training dataset generation based on different automated unsupervised learning is designed which make the program more powerful than depending on a clean training dataset for another learning scheme.

A: The unsupervised methods including DeepCryoPicker are designed to automatically pick particles without human intervention. However, the problem of the unsupervised methods is that the picking results are usually not reliable. To demonstrate this problem, we used a Fab sample (the antigen binding fragment), which is a small protein and has high concentration on micrographs. DeepCryoPicker failed to pick particles on this dataset. On the contrary, although not trained on similar datasets, EPicker can initially use a general model to pick part of the particles. After roughly removing some bad particles, the remaining sparse particles are enough to train a new model to pick all rest particles. As EPicker can perform training in a continual manner, the knowledge of the Fab will be permanently accumulated into the model, and ready for future use. Therefore, EPicker with continual learning strategy is more reliable and can be used to

enhance the automated cryoEM pipeline with continuous data flow.

The following (**Figure R3.1**) shows the comparison of EPicker with DeepCryoPicker and other programs. More discussion has been added in the **Supplementary Figure 6** of the revised manuscript.

Figure R3.1. Comparison on different software. Because the size of Fab is very small relative to the whole micrograph, in order to improve the visualization, we just show a small region of a raw micrograph. A Fab dataset containing 179 micrographs was used to evaluate different software. A ribosome micrograph was also used. The Fab rarely appears in public databases, and hence is a good “unknown” sample for most published software. In contrast, the ribosome dataset is contained in most of public databases, and thus is a “known” sample for the published software. **a ~ c)** Typical picking results on the ribosome and Fab micrographs of EPicker, crYOLO and TOPAZ, respectively. Here we used general models associated with three software. The general model of EPicker was trained on 50 heterogeneous datasets excluding Fab (please see **Supplementary Table 8** in the revised manuscript). The general model of crYOLO was trained on a combination of 53 datasets. And the number of datasets used to train the general model of TOPAZ is unknown. All these three models show ideal performance on the ribosome, but not ideal on the Fab. **d~e)** Picking results of DRPNNet and DeepCryoPicker on Fab. Both software employed unsupervised-learning algorithms and failed to pick Fabs. No further tests can be performed since a training procedure is not

available. **f)** Ten selected 2D classes and corresponding particle annotations of Fab. To generate a training dataset, the initial picking results of EPicker (shown in **a**) on 8 micrographs in the Fab dataset were filtered by THUNDER 2D classification. In the ten classes, 6562 particles showing obvious features of the Fab were selected as training annotations, and then were used to build the training dataset for EPicker, crYOLO and TOPAZ. The ten class averages were used as the templates for RELION. **g ~ i)** Typical picking results on the ribosome and Fab micrographs of EPicker, crYOLO and TOPAZ, respectively, after loading the corresponding general models and training on the newly built Fab dataset. The continual learning was used for EPicker and the fine-tuning was used for crYOLO and TOPAZ. All three software performed as expected on the Fab. For crYOLO and TOPAZ, the performance on the ribosome dataset was not maintained, indicated by lots of missed picking on the ribosome. While the picking results of EPicker were similar to those before the training on Fab. **j)** A typical picking result of RELION based on template matching using the ten class averages. RELION cannot avoid picking ice particles and hence did not perform as well as the other three deep-learning-based software. **k ~ n)** The results of 2D classification on the picking results of EPicker, crYOLO, TOPAZ and RELION, corresponding to the processing in **g ~ j**. All software picked ~ 600,000 particles (details shown on the bottom of each figure). After empirically selecting classes well centered and with secondary-structure details, EPicker and RELION output more good particles (~390,000) than the other two software.

6-Paper is focusing on comparing the results with the other CNNs or deep learning scheme and missing comparing the picking results with the some state-of-the-art approaches in the cryo-em filed such as RELION 3.1 and EMAN 2.31.

A: Thanks for the suggestion. We compared EPicker with more programs, please see the results shown above (**Figure R3.1**). We didn't compare with EMAN, because EMAN uses a manual picking algorithm which is strongly dependent on user's operations.

7-A key feature of the design network is based on using an raining example during the training and is considered as a reference for old features, which preserves the old knowledge and that will be good is the training dataset that been selected be in the same training domain such as shape and size, however, if the training sample differs from dataset to another that will affect the whole training scheme.

A: EPicker has shown excellent ability of generalization in accumulating features with significant differences in our experiments. Particles used in our experiments are heterogenous in shapes and sizes (see **Table R3.1**). For example, β -galactosidase (EMPIAR-10017) is very different from the Nodavirus (EMPIAR-10203) in both size and shape. However, the differences between them only slightly affect the whole training scheme. As it can be seen from the second row of **Table R3.2**, when EPicker was incrementally trained on the β -galactosidase dataset, it did not lose its ability to pick Nodavirus.

We made a semi-quantitative analysis to give a deeper insight into the behavior of the continual learning when particles with different features are added.

Adding more training datasets means merging more features into an old model trained on the old datasets. The dissimilarity of features between the old and new datasets may influence the effectiveness of accumulating features (knowledge), which is indicated by forgetting some old features. To evaluate this influence, we defined the complexity of a new dataset as how the features in the new dataset match the features

in the old datasets. The higher the complexity, the greater the differences. Then we sought for a relationship between complexity and forgetting rate using 10 datasets (**Table R3.1**).

Table R3.1. Illustration of images in different training datasets after local enlargement.

E10089 (TcdA1)	E10146 (Apoferitin)	E10028 (80S ribosome)	E10203 (Nodavirus)	E10025 (20S Proteasome)
				E10017 (Beta-galactosidase)	E10097 (Influenza hemagglutinin trimer)	E10075 (Phage MS2)	E10081 (HCN1 ion channel)	E10228 (Phosphodiesterase 6)
				
The complexity and the forgetting rate metrics are formulated as follows:

- **The complexity.**

To assess the complexity of a new dataset, we use an old general model to pick new particles. The complexity of the new dataset is inversely proportional to the picking accuracy, and is empirically defined as

$$C = \frac{100}{10^{(AP+AR)}}$$

where $C \in [1,100]$ is the complexity of the new dataset, AP and AR refer to the average precision and average recall when directly using the old general model to pick new particles. When the features of new particles are significantly different from those in the old datasets, the picking may fail and lead to low AP and AR, consequently, the high complexity.

- **The forgetting rate.**

The reduction in AP and AR indicates forgetting. We defined the forgetting rate for AP and AR, respectively, as the average reduction in AP and AR on all old datasets before and after training on a new dataset.

For the evaluation, we still used the ten datasets (**Table R3.1**). The AP/AR values used for calculating the complexity and the forgetting rate are shown in the following **Table R3.2**.

Table R3.2. Evaluation of particle picking (AP/AR) under different training modes. The 10 datasets are indicated by D1 ~ D10, the corresponding exemplar dataset are indicated by E1 ~ E10. The joint training (JT) and continual learning (CL) are tested with different datasets. The cells with grey background indicate that the

corresponding dataset is unseen for the old model. The cells with underline values indicate that the corresponding dataset is newly added for the model.

Name	E10089	E10146	E10028	E10203	E10025	E10017	E10097	E10075	E10081	E10228
NO.	(1)	(2)	(3)	(4)	(5)	(6)	(7)	(8)	(9)	(10)
JT(D ₁ ~D ₅)	97.5/99.1	96.4/97.2	96.7/97.5	93.1/98.9	92.0/96.6	19.8/50.2	9.7/64.9	95.1/98.7	80.9/92.3	26.5/77.6
CL	96.4/99.2	94.9/97.2	96.4/97.7	90.2/98.3	90.3/95.5	96.2/98.5	66.5/97.2	92.2/96.7	84.4/95.2	55.9/97.0
(E ₁ ~E _t +	96.1/98.4	96.2/97.6	96.6/97.3	89.4/97.7	90.2/95.7	96.4/98.8	93.9/98.5	92.2/96.4	88.6/96.9	84.1/97.4
D _{t+1})	96.1/98.8	96.7/97.9	96.6/97.3	92.6/98.9	89.9/95.3	96.4/99.0	91.9/96.8	95.7/96.7	80.1/95.0	83.4/96.4
t = 5 ~ 9.	96.4/98.8	96.7/97.7	96.7/97.3	91.9/98.9	89.7/95.8	95.8/98.3	92.8/97.5	96.4/97.1	94.9/97.7	80.8/96.8
	96.4/98.8	96.8/97.6	96.6/97.3	90.1/98.9	89.6/95.6	94.6/97.9	91.8/97.5	95.8/96.7	93.1/96.9	91.8/97.2

A plot is calculated based on the data in **Table R3.2**, and shown in the following **Figure R3.2**.

Figure R3.2. Relation between complexity of new dataset and forgetting rate on the old datasets. The complexity (yellow) and the forgetting rate of AP (blue) and AR (light blue) for sequentially added new dataset from D6 to D10.

It can be seen from **Figure R3.2** that the complexity has strong correlation with the forgetting rate. Dataset 6 (Beta-galactosidase in **Table R3.1**) has the highest complexity, which does have very different features from those of dataset 1 ~ 5 (see the first row in **Table R3.1**). While the high complexity of dataset 6 causes a forgetting rate up to 1.5%, the forgetting is still minor comparing with the absolute AP value of ~90%. The low complexities of dataset 7 ~ 10 just cause a subtle forgetting, and even improve the performance of picking on the old datasets (Dataset 7 ~ 9, indicated by the negative forgetting rates).

In summary, adding new dataset with different features (complexity) won't cause significant forgetting. Meantime, adding datasets with low complexity sometimes can improve the performance on the model. Therefore, with larger and larger training datasets involved, EPicker can maintaining the performance of the model on the old datasets.

8-We need to see some other measurement criteria such as AUC curve, and others

such as precision, recall, the accuracy, lost and training plot ect. which is the most important fact to judge the design network.

A: Thanks for the suggestion. We have tested other metrics, and found them show the similar tendency as the mAP/mAR. Therefore, we didn't include these new metrics in the manuscript. We prepared the required data as follows. In addition to the mAP/mAR, we also plotted the variation curve of training loss (See **Figure R3.3**), and calculated the Area Under Curve (AUC) and F1-score as performance metrics (see **Table R3.3**). AUC is defined as the area enclosed by the coordinate axis under the ROC curve. F1-score is the harmonic average of precision value and recall value. All these new metrics show the similar tendency as mAP/mAR, which confirms the conclusion in the submitted manuscript. Hence, we concluded that mAR/mAP is sufficient to reflect the particle picking performance of EPicker. The details are as shown in **Figure R3.3**.

Figure R3.3. Metrics to evaluate the continually learning on 5 datasets. We first loaded an initial model, which was trained on micrographs from five EMPIAR datasets: 1–5. Then, EMPIAR-10017 was used for continual learning based on the initial model. **a)** Variation curve of training loss. With the increase of training time, the training loss decreased gradually and finally tended to converge. **b)** Variation curves of AP and AR on the training dataset. With the increase of training time, AP and AR values on the training dataset increased and finally converged. **c)** Variation curves of AP and AR on the test dataset. With the increase of training time, AP and AR values on the test dataset increased and finally converged. **d)** Precision-Recall (PR) curve on the test dataset of EMPIAR-10017. **e)** Receiver Operating Characteristic (ROC) curve on the test dataset of EMPIAR-10017.

Table R3.3. Evaluation results of the continual learning model in EPicker. We first loaded an initial model, which was trained on micrographs from five EMPIAR datasets: 1–5. Then, EMPIAR-10017 was used for continual learning based on the initial model. The AP, AR, F1-score and AUC for all the test datasets are recorded.

Dataset	AP	AR	F1-score	AUC
E10089	96.4	99.2	97.8	97.4
E10146	94.9	97.2	96.0	95.5

E10028	96.4	97.7	97.0	97.3
E10203	90.2	98.3	94.1	96.3
E10025	90.3	95.5	92.8	93.5
E10017	96.2	98.5	97.3	96.3

9-Consuming time for training and testing is missing.

A: Table R3.4 lists the training time and picking time of different methods. We joint trained all the models on a combination of 10 datasets, including 100 micrographs and 19300 annotated particles. Picking time recorded the time required for the model to pick particles in one micrograph.

Table R3.4 Comparison of training time and picking time of different software. To evaluate the training process, we compared different software on a combination of 10 datasets (used in Table 1 of the submitted manuscript), including 100 micrographs and 19300 annotated particles. All the experiments were performed on a single GPU (RTX 2080Ti). The time of joint training on 10 datasets was measured for crYOLO, Topaz and EPicker. For EPicker, the time for continual learning was measured by adding 10th dataset to a general model trained on the previous 9 datasets. RELION wasn't included in the comparison. To evaluate the picking process, the average time for picking a single micrograph was recorded.

Software	Training time (100 micrographs)	Picking time (1 micrograph)
TOPAZ (Joint Training)	30 min	0.5 s
crYOLO (Joint Training)	56 min	0.3 s
DRPNet	×	3 s
DeepCryoPicker	×	30 s
EPicker (Joint Training)	50 min	0.3 s
EPicker (Continual Learning)	20 min	0.3 s

10-We need to see the time consuming for the picking part.

A: The time used for picking on one micrograph is 0.2~0.3 s on a GTX2080Ti GPU. We have added the time information in the revised manuscript.

11-is the testing dataset was external which means does the program uses some cryo-EM images for an external data that the deep learning approach did not see during the training.

A: In all our experiments, we selected 15 micrographs from each dataset, of which 10 were used to build the training dataset and 5 were used to build the test dataset. Hence, the test dataset and the training dataset are irrelevant, which can better evaluate the generalization ability of the proposed method.

12-The most important fact in the 3D density map by the single particle picking is the density map resolution, we need to see the FSC curve at the .134 to check the final resolution output the program.

A: We don't agree that FSC is the best indicator to evaluate particle picking, because

too many factors can influence the FSC. The particle picking is performed in the early stage of the cryoEM workflow. Before computing the final 3D volume, a lot of procedures are conducted on the picking results, such as 2D and 3D classification. In many real cases, only 5~10% picked particles are used in the final reconstruction. More importantly, a high-resolution 3D reconstruction requires homogeneous particles, but the users usually hope to pick all heterogeneous particles to study the dynamic conformation. Therefore, the resolution of 3D reconstruction is not directly correlated to the performance of the particle picking. Using AP/AR metric or 2D classification to evaluate a particle picking algorithm is more feasible.

13- You have to compare that as well with the other software's in the filed such as RELION 3.1, and EMAN 2.31.

A: Thanks for the suggestion. In the revised manuscript we compared the proposed EPicker with other widely used single particle picking framework, including supervised methods: RELION, crYOLO, Topaz, and unsupervised methods: DRPNet, DeepCryoPicker. The results are shown in above **Figure R3.1** (We didn't compare EPicker with EMAN, because EMAN is based on a kind of manual picking algorithm which is strongly depended on user's operation.)

Reviewers' Comments:

Reviewer #1:

Remarks to the Author:

Most of my previous questions have been addressed. I agree with the publication.

Reviewer #2:

Remarks to the Author:

I commend the authors for the several improvements performed in the manuscript.

I still have two minor comments:

The code is now available and accompanied by proper documentation and tutorials. Although Tutorials 4 and 5 are still missing. I would strongly recommend the authors to upload it before publication.

I understand that some hyper parameters (λ_{off} , λ_{size}) were chosen as recommended by the CenterNet authors in their original publication. But as the centerNet publication was not originally assessed on Cryo-EM images, I still believe that it would be desirable to verify that these hyper parameters are still appropriate for continual learning on Cryo-EM data.

Reviewer #3:

Remarks to the Author:

The main contribution of this manuscript is based on designing Deep Learning algorithms on unseen datasets with different features that are often unpredictable, however recently many papers have been published with a very significant approach that has been tested on unseen data. Also, the proposed system reports an exemplar-based continual learning approach, which can accumulate knowledge from the new dataset into the model by training an existing model on only a few new samples without catastrophic forgetting of old knowledge, implemented in a new program called EPicker.

REVIEWERS' COMMENTS

Reviewer #1 (Remarks to the Author):

Most of my previous questions have been addressed. I agree with the publication.

A: Thanks for your time and efforts.

Reviewer #2 (Remarks to the Author):

I commend the authors for the several improvements performed in the manuscript.

I still have two minor comments:

The code is now available and accompanied by proper documentation and tutorials. Although Tutorials 4 and 5 are still missing. I would strongly recommend the authors to upload it before publication.

I understand that some hyper parameters (λ_{off} , λ_{size}) were chosen as recommended by the CenterNet authors in their original publication. But as the centerNet publication was not originally assessed on Cryo-EM images, I still believe that it would be desirable to verify that these hyper parameters are still appropriate for continual learning on Cryo-EM data.

A: Thanks for the suggestion. We have uploaded tutorials 4 and 5. The website is now available through <http://www.thuem.net/software/epicker/tutorial.html>.

We have added experiments into the revised manuscript for the setting of hyper-parameters λ_{off} and λ_{size} . The evaluation results are following.

The loss function of EPicker is defined as

$$L_{Total} = L_k + \lambda_{off} * L_{off} + \lambda_{size} * L_{size} + \lambda_d * L_{Distill} + \lambda_r * L_{Reg}$$

The first two hyper-parameters control the weight of prediction precision of particle size and local offset. When the weighting factor λ_{off} is larger, the model is more inclined to penalize the prediction loss term of particle center. On the contrary, when the weighting factor λ_{size} is larger, the model tends to predict the particle size more accurately. Therefore, we performed test for different settings (see **Table R1**). The different settings seems leading to very closed performance. Finally, we chose the parameter set $\lambda_{off} = 1$ and $\lambda_{size} = 0.1$ with the best performance, which are the same as that used in original CenterNet.

Table R1. Comparison of different weighting factors for particle detection. Liposome dataset was used for training and test. EPicker predicted the size and the position of liposome simultaneously. EPicker chose the weighting factors $\lambda_{off} = 1$ and $\lambda_{size} = 0.1$, which obtained the highest mAP and mAR.

Strategy	$\lambda_{off} = 1, \lambda_{size} = 1$	$\lambda_{off} = 1, \lambda_{size} = 0.1$	$\lambda_{off} = 0.1, \lambda_{size} = 1$	$\lambda_{off} = 0.1, \lambda_{size} = 0.1$
Liposome	91.7/99.4	92.8/99.4	92.0/99.4	90.6/98.8

Reviewer #3 (Remarks to the Author):

The main contribution of this manuscript is based on designing Deep Learning algorithms on unseen datasets with different features that are often unpredictable, however recently many papers have been published with a very significant approach that has been tested on unseen data. Also, the proposed system reports an exemplar-based continual learning approach, which can accumulate knowledge from the new dataset into the model by training an existing model on only a few new samples without catastrophic forgetting of old knowledge, implemented in a new program called EPicker.

A: Thanks for your time and efforts. As we tested in the manuscript, all programs we tested is not reliable on new dataset with significantly different features from those they have seen. This is reasonable for the deep-learning method based on CNNs. Therefore, a further training will be always necessary for improving either the accuracy or reliability, which is EPicker aims to.